\

# A processive rotary mechanism couples substrate unfolding and proteolysis in the ClpXP degradation machinery

Zev A Ripstein[1,2†*], Siavash Vahidi[1,2,3,4†*], Walid A Houry[1,4],
John L Rubinstein[1,2,5*], Lewis E Kay[1,2,3,4*]

[1]Department of Biochemistry, University of Toronto, Toronto, Canada; [2]The Hospital for Sick Children Research Institute, Toronto, Canada; [3]Department of Molecular Genetics, University of Toronto, Toronto, Canada; [4]Department of Chemistry, University of Toronto, Toronto, Canada; [5]Department of Medical Biophysics, University of Toronto, Toronto, Canada

**Abstract** The ClpXP degradation machine consists of a hexameric AAA+ unfoldase (ClpX) and a pair of heptameric serine protease rings (ClpP) that unfold, translocate, and subsequently degrade client proteins. ClpXP is an important target for drug development against infectious diseases. Although structures are available for isolated ClpX and ClpP rings, it remains unknown how symmetry mismatched ClpX and ClpP work in tandem for processive substrate translocation into the ClpP proteolytic chamber. Here, we present cryo-EM structures of the substrate-bound ClpXP complex from *Neisseria meningitidis* at 2.3 to 3.3 Å resolution. The structures allow development of a model in which the sequential hydrolysis of ATP is coupled to motions of ClpX loops that lead to directional substrate translocation and ClpX rotation relative to ClpP. Our data add to the growing body of evidence that AAA+ molecular machines generate translocating forces by a common mechanism.

**\*For correspondence:**
zevripstein@gmail.com (ZAR);
svahidi@pound.med.utoronto.ca
(SV);
john.rubinstein@sickkids.ca (JLR);
kay@pound.med.utoronto.ca
(LEK)

†These authors contributed
equally to this work

## Introduction

Protein degradation plays a central role in cellular physiology, regulating the timing of cell division, controlling stress responses, and ensuring the timely removal of damaged or aberrantly folded proteins (*Olivares et al., 2016*; *Goldberg, 2003*). The ClpXP system is central to protein degradation in bacteria and in mitochondria, where it has recently emerged as a therapeutic target against malignancies (*Ishizawa et al., 2019*), and is a key regulator of cellular homeostasis, pathogenesis, and intracellular parasitism (*Frees et al., 2014*; *Pickart and Cohen, 2004*; *Bhandari et al., 2018*). It also serves as a model for understanding the structure and function of other ATP-dependant proteolytic systems. The ClpXP holoenzyme is composed of a hexameric AAA+ (ATPases Associated with diverse cellular Activities) unfoldase (ClpX) and a tetradecameric serine protease consisting of two heptameric rings (ClpP). Together, the ClpXP complex acts to unfold and degrade protein substrates (*Figure 1A*; *Baker and Sauer, 2012*). ClpXP function is critical in many bacteria, and consequently small molecules that disrupt either the protease (e.g. β-lactones [*Böttcher and Sieber, 2008*], phenyl esters (*Hackl et al., 2015*; *Lakemeyer et al., 2019*) or the AAA+ unfoldase (e.g. ecumicin (*Gao et al., 2015*), lassomycin (*Gavrish et al., 2014*), rufomycin (*Choules et al., 2019*), dihydrothiazepines (*Fetzer et al., 2017*), or mimic the interaction between the two (e.g. acyldepsipeptides [ADEPs] (*Kirstein et al., 2009*; *Gersch et al., 2015*; *Wong et al., 2018*) can kill cells, establishing ClpXP as a target for novel therapeutics against a range of different infectious diseases and cancers (*Culp and Wright, 2017*; *Malik and Brötz-Oesterhelt, 2017*; *Raju et al., 2012*; *Goard and Schimmer, 2014*; *Wong and Houry, 2019*).

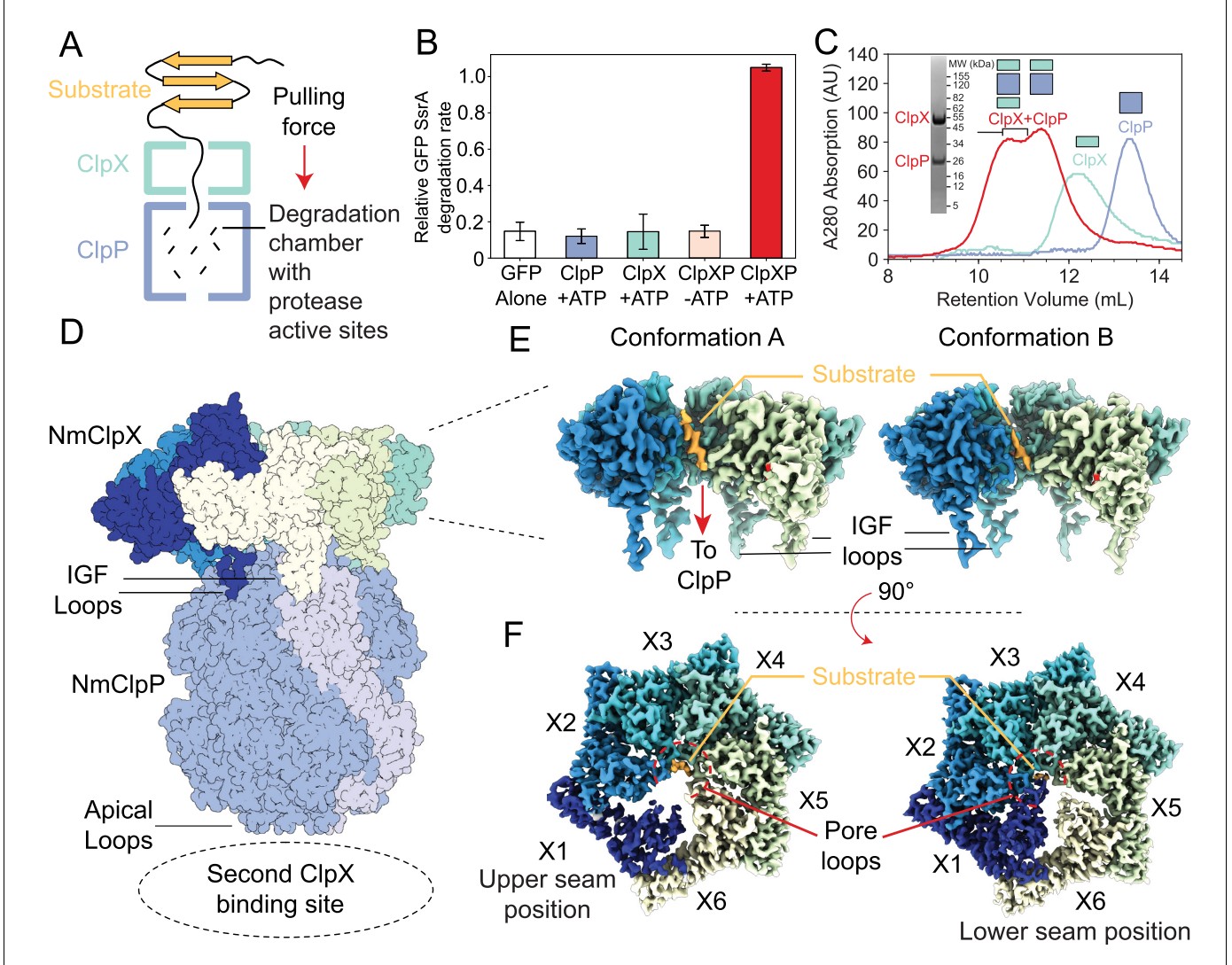

**Figure 1.** Functional and structural characterization of ClpXP from *N. meningitidis*. (**A**) Schematic representation of the ClpXP degradation machinery. The overall positions of the substrate (orange), ClpX unfoldase (green), and the ClpP protease (blue) are shown; (**B**) GFP-SsrA degradation by ClpXP is ATP-dependent. The degradation rate of GFP-SsrA is monitored by measuring loss of fluorescence. The components (ClpX, ClpP, ATP) included for each measurement are denoted on the plot. All measurements included GFP-SsrA and were performed in triplicate on WT ClpXP; (**C**) ClpXP complex formation monitored by size exclusion chromatography (SEC). SEC profiles of isolated ClpP (blue trace) and ClpX (green trace) are consistent with their expected molecular weights. SEC profile of a 2:1 ClpX$_6$:ClpP$_{14}$ mixture (red trace) incubated for 10 min in the presence of MgATP shows the formation of doubly- and singly-capped ClpXP complexes. The running buffer for all traces contained 2 mM MgATP. SDS-PAGE gel of SEC fractions (fractions within the region delineated by black bracket) shows that they contain both ClpP and ClpX; (**D**) Overall architecture of the ClpXP complex as established by cryo-EM. The ClpP double ring component is shown in dark blue, with the exception of a pair of opposing protomers shown in light blue shade to delineate the IGF loop binding site that is located at the interface between protomers. Cutaway side (**E**) and top (**F**) views of Conformations A and B of ClpX resulting from focused classification and local refinement. The IGF loops that bind to ClpP, and the presence of substrate (orange) in the axial channel of ClpX, are highlighted. Each of the ClpX protomers is labeled. In panel (**F**), the engagement of the substrate (orange) with five of the six pore loops is indicated (red semi-circle).

The online version of this article includes the following figure supplement(s) for figure 1:

**Figure supplement 1.** Sequence map of NmClpX and SEC analysis of substrate binding by NmClpXP.

**Figure supplement 2.** Cryo-EM image processing.

**Figure supplement 3.** Cryo-EM map validation.

**Figure supplement 4.** Examples of regions of the atomic models built into the experimental cryo-EM maps.

**Figure supplement 5.** Experimental density maps and models for the substrate pore-1 loop interaction interface.

X-ray crystallographic studies of ClpP alone reveal a barrel-like structure with the protease rings stacked coaxially to form an enclosed degradation chamber, akin to the 20S proteasome (*Förster et al., 2013*; *Kish-Trier and Hill, 2013*), that sequesters the fourteen Ser-His-Asp catalytic triads (*Wang et al., 1997*; *Liu et al., 2014*). Early negative-stain electron microscopy showed that one or two ClpX particles can bind co-axially to ClpP, creating a continuous central channel for substrates from ClpX to the degradation chamber of ClpP (*Ortega et al., 2002*; *Ortega et al., 2000*; *Kessel et al., 1995*). Biochemical and X-ray crystallographic studies show that the axial entrances to the degradation chamber are gated by flexible N-terminal loops of ClpP that open upon the binding of ClpX or ADEPs, allowing substrates into the degradation chamber (*Lee et al., 2010*; *Li et al., 2010*; *Effantin et al., 2010*). Binding of ClpX and ClpP is mediated by loops containing an Ile/Leu-Gly-Phe motif (I/LGF loops) on ClpX that dock into hydrophobic pockets on the apical interfaces of ClpP subunit pairs (*Kim et al., 2001*; *Joshi et al., 2004*; *Amor et al., 2019*). N-terminal loops on ClpP also transiently interact with ClpX (*Joshi et al., 2004*; *Gribun et al., 2005*; *Jennings et al., 2008*). However, the precise nature of these interactions and how they allow coordination of the activities of ClpX and ClpP remains unclear.

ClpX uses energy released by ATP hydrolysis to create a pulling force that unravels folded protein domains for translocation into the degradation chamber of ClpP (*Baker and Sauer, 2012*). This unfolding activity relies on a series of conserved loops in ClpX. These loops include the 'RKH' loops that surround the ClpX entrance pore and recognize substrates for proteolysis (*Martin et al., 2008a*), and a pair of 'pore loops' in each protomer, termed pore-1 loop (GYVG in *Neisseria meningitidis* ClpX, NmClpX) and pore-2 loop (RDV in NmClpX) (*Figure 1—figure supplement 1A*) that line the axial channel of the ClpX ring (*Martin et al., 2008b*). Bulky aromatic and aliphatic side chains of the pore loops transmit the pulling force to the substrate, leading to its unfolding. The mechanism by which some AAA+ ATPases convert energy from ATP hydrolysis into a mechanical force that unfolds substrates was established recently (*Monroe et al., 2017*; *Yu et al., 2018*; *De la Peña et al., 2018*; *Shin et al., 2019*; *Ripstein et al., 2017*; *Gates et al., 2017*), and structures of ClpP and ClpX in isolation have been determined (*Ishizawa et al., 2019*; *Li et al., 2016*; *Geiger et al., 2011*; *Glynn et al., 2009*). However, the absence of structures showing ClpXP in the process of unfolding substrate has prevented an understanding of the conformational changes that are necessary for substrate engagement, unfolding, and translocation in this system. ClpXP must undergo hundreds of ATP hydrolysis events to unfold and degrade a single protein, likely via a process involving hydrolysis of one nucleotide at a time, but the lack of structural information for a ClpXP holoenzyme has limited understanding of how unfolding and degradation are coupled. For example, it is not clear how the symmetry mismatch (*Bewley et al., 2006*; *Beuron et al., 1998*; *Majumder et al., 2019*) between the pseudo-symmetric hexameric ClpX and the seven-fold symmetric ClpP affects both the binding and interaction of these two components.

Here, we present cryo-EM structures of the ClpXP holoenzyme from the gram-negative pathogen *Neisseria meningitidis* engaged with a protein substrate. The ClpXP portion of the structure is at 2.3 to 3.3 Å resolution, allowing construction of a nearly complete atomic model of the complex. Our findings show how ClpX grips and translocates substrates into the degradation chamber through a cycle of ATP hydrolysis events involving both concerted motions of pore loops along the substrate, as well as motions of ClpX on the apical surface of ClpP. The data lead to a model for interactions between symmetry-mismatched ClpX and ClpP that enable continuous degradation of substrates as ClpX rotates relative to ClpP.

## Results

### ATP-dependent binding and degradation of GFP-SsrA by ClpXP

ClpX and ClpP from *N. meningitidis* (Nm) were expressed separately in *E. coli* before purification with metal affinity chromatography and size exclusion chromatography (SEC). Degradation assays using green fluorescent protein bearing an eleven-residue SsrA tag (GFP-SsrA) (*Ripstein et al., 2017*) were performed to ensure that the heterologously expressed NmClpX and NmClpP proteins, when mixed, have the well-established activity of the ClpXP holoenzyme. The presence of ClpP or ClpX alone led to no loss of GFP-SsrA fluorescence relative to background bleaching in the GFP-alone control assay. ClpX, ClpP, and ATP were all necessary to degrade GFP-SsrA (*Figure 1B*),

confirming that degradation of GFP-SsrA by ClpXP is ATP-dependent. As shown below and in previous work (*Joshi et al., 2004*; *Hersch et al., 2005*; *Grimaud et al., 1998*; *Jones et al., 1998*), the presence of MgATP is required for tight binding between ClpX and ClpP. Therefore, for further structural studies, an E185Q Walker B mutant of NmClpX (ClpX-WB) was used to slow ATP hydrolysis and prolong the lifetime of the ClpXP complex. SEC was performed to probe the oligomeric state of each component and the formation of the holoenzyme, confirming that ClpP behaves as an oligomeric species, consistent with its well-known tetradecameric architecture (*Figure 1C* – blue trace). ClpX-WB eluted as a hexamer in the presence of 2 mM MgATP (*Figure 1C* – green trace). Incubation of ClpX-WB and ClpP in the presence of MgATP resulted in a pair of higher molecular weight SEC peaks that contain both ClpX and ClpP (*Figure 1C* – red trace and SDS-PAGE gel insert), presumably corresponding to singly and doubly capped ClpXP complexes, showing that the *N. meningitidis* ClpX:ClpP complex can be readily reconstituted in vitro.

To isolate a substrate-bound holoenzyme, ClpXP was incubated with GFP-SsrA in the presence of MgATP. SEC fractions corresponding to doubly capped ClpXP bound to GFP-SsrA (*Figure 1—figure supplement 1B* - denoted with a *) were applied to specimen grids and vitrified for cryo-EM analysis. Visual inspection of the micrographs, as well as 2D classification of particle images, showed doubly capped complexes (*Figure 1—figure supplement 2*). However, 2D classification showed multiple classes where the two ClpX rings have lower density than the ClpP region, and other classes with one ClpX ring having strong density and the other having weak density. An *ab initio* 3D map showed ClpP along with strong density for one ClpX ring and only weak and fragmented density for the other (*Figure 1—figure supplement 2C*). Inspection of the map revealed that the ClpX ring is offset and tilted relative to the ClpP symmetry axis (*Figure 1D*, *Video 1*). The poor density for the second ClpX hexamer is likely due to a lack of correlation between binding offsets of the two ClpX rings that leads to incoherent averaging of one of the ClpX rings in the map. The map of the intact complex was refined to an overall resolution of 2.8 Å (*Figure 1—figure supplement 2C*); however, the density for the better-resolved ClpX ring remained fragmented. Further local refinement and classification in cryoSPARC (*Punjani et al., 2017*) improved the density for ClpX and revealed two distinct conformations, at 3.3 Å and 2.9 Å resolution (*Figure 1—figure supplement 2*), hereafter referred to as Conformations A and B, respectively. Focused refinement of ClpP with D7 symmetry enforced led to a map at 2.3 Å resolution (*Figure 1—figure supplement 2*).

## Overall architecture of NmClpXP

Atomic models built into the cryo-EM maps showed that in both Conformations A and B, ClpX is positioned with an offset and is tilted relative to the seven-fold symmetry axis of ClpP (*Figure 1D*, *Video 1*). Nevertheless, a continuous channel for substrate spans the ClpX and ClpP rings. ClpP is in its active extended conformation (*Goodreid et al., 2016*), and closely resembles a published ADEP-bound crystal structure with an all atom RMSD of 1.18 Å for the tetradecamer (pairwise comparison of the chains gives values from 1.12 to 1.22 Å with a standard deviation from the mean of 0.03 Å) (*Goodreid et al., 2016*). Even though the full-length constructs of ClpX included the zinc binding domains (residues 1–62), no density was found for these domains in either conformation, likely due to their flexibility.

Conformations A and B of ClpX both contained additional density along their axial channels, adjacent to the pore loops, likely from the GFP-SsrA substrate that co-eluted with ClpXP from the SEC column (*Figure 1E and F* - orange) (*Figure 1—figure supplement 1B*). As the Walker B mutant has low unfolding activity (*Martin et al., 2008c*), the density observed

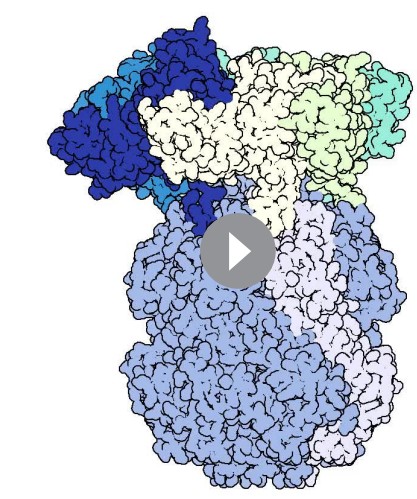

**Video 1.** Overview of the substrate-bound ClpXP complex (Conformation B).
https://elifesciences.org/articles/52158#video1

likely corresponds to the unstructured region of the degron tag. The substrate enters ClpX at an angle of ~15° relative to the ClpP symmetry axis and is gripped by five of the six ClpX protomers arranged in a right-handed spiral, akin to other AAA+ ATPases in their substrate-bound conformations (*Monroe et al., 2017*; *Yu et al., 2018*; *De la Peña et al., 2018*; *Shin et al., 2019*; *Ripstein et al., 2017*; *Gates et al., 2017*). The sixth protomer forms a disengaged 'seam' that bridges protomers at the beginning and end of the spiral. In the description that follows we have labeled the protomers X1 through X6 corresponding to their position in the spiral (*Figure 1F*), and consider the configuration in which X1 is the disengaged 'seam' protomer of Conformation A.

The main differences between Conformations A and B are within the protomer that adopts the bridging seam position, and where the seam is located relative to ClpP. In Conformation A, protomer X1 is disengaged from substrate and in the seam position (*Figure 1F* left - dark blue) while in Conformation B protomer X6 is disengaged from substrate and in the seam position (*Figure 1F* right - yellow). However, these two seam positions are not equivalent: in Conformation A the seam is further from the apical surface of ClpP in what we designate as the 'upper seam' (US) position (*Figure 1D,F* - dark blue), while in Conformation B, the seam is closer to ClpP in the 'lower seam' (LS) position (*Figure 1D,F* - yellow).

## ClpX IGF loops

In both Conformations A and B of ClpXP, there is clear density for most of the IGF loops that extend from the ClpX ring toward the apical surface of ClpP (*Figure 2A*). In Conformation A, clear density is observed for only five of the six IGF loops, with weak density belonging to the IGF loop of the X6 protomer (white dotted line in *Figure 2B* left; *Figure 2—figure supplement 1*). In Conformation B, the density of the X6 IGF loop is significantly stronger than in Conformation A, so that all six of the IGF loops are accounted for (residues 264–274, *Figure 2B*, right; *Figure 2C*; *Figure 2—figure supplement 1*), with the loops binding six of the seven pockets on ClpP. In both Conformations A and B, the empty ClpP-binding pocket is located between the X5 and X6 protomers of ClpX (*Figure 2B*). The lack of density for the IGF loop belonging to protomer X6 in Conformation A (*Figure 2—figure supplement 1*) is likely due to conformational flexibility and sub-stoichiometric binding of this IGF loop into the ClpP-binding pocket.

All six IGF loops in ClpX adopt different configurations in Conformations A and B (*Figure 2—figure supplement 1*). The ClpX residues immediately following where the IGF loops contact ClpP (residues 275–280) show weak or no density, likely indicating flexibility. This property allows the IGF loops to accommodate movement of the ClpX protomers relative to ClpP (*Figure 2D–F*), analogous to a set of springs. For instance, in Conformation A the X1 protomer is closer to the ClpP surface than in Conformation B and its IGF loop is thus compact in this state. In contrast, in Conformation B the loop is fully extended, allowing the X1 protomer to reach upwards toward the top of the spiral to engage the substrate (*Figure 2F*; *Figure 2—figure supplement 1*). In this way, the flexibility of the IGF loops accommodates the spiral arrangement of ClpX necessary to bind and unfold substrate, while maintaining contact with the planar ring of ClpP. Additionally, the IGF loops allow the X2 protomer to move laterally away from the ClpP pore while the X4 and X5 protomers move toward it to form a structure in which the ClpP and ClpX rings are displaced with respect to each other (*Figure 2D*-red arrow; *Figure 2E*).

## ClpP N-terminal gates

Degradation assays (*Effantin et al., 2010*; *Gribun et al., 2005*; *Martin et al., 2007*; *Ortega et al., 2004*; *Singh et al., 2000*) show that ClpX allows substrate access to the degradation chamber of ClpP, suggesting that ClpX binding opens the N-terminal gates of ClpP. Indeed, in both Conformations A and B the gates are in the 'up' state, forming ordered β-hairpins (*Figure 2G*; *Figure 2—figure supplement 2*). These β-hairpins closely resemble the conformation seen in the ADEP-bound structures of NmClpP (*Goodreid et al., 2016*) and are notably different from the disordered conformations observed in NmClpP when it is not bound to either ClpX or ADEP (*Figure 2—figure supplement 2B*; *Mabanglo et al., 2019*). By forming ordered gates, ClpX binding creates a wide entrance pore with a diameter of ~23 Å for substrates to pass through into the ClpP degradation chamber (*Figure 2G*, left; *Figure 2—figure supplement 2*). While it has long been suspected that the

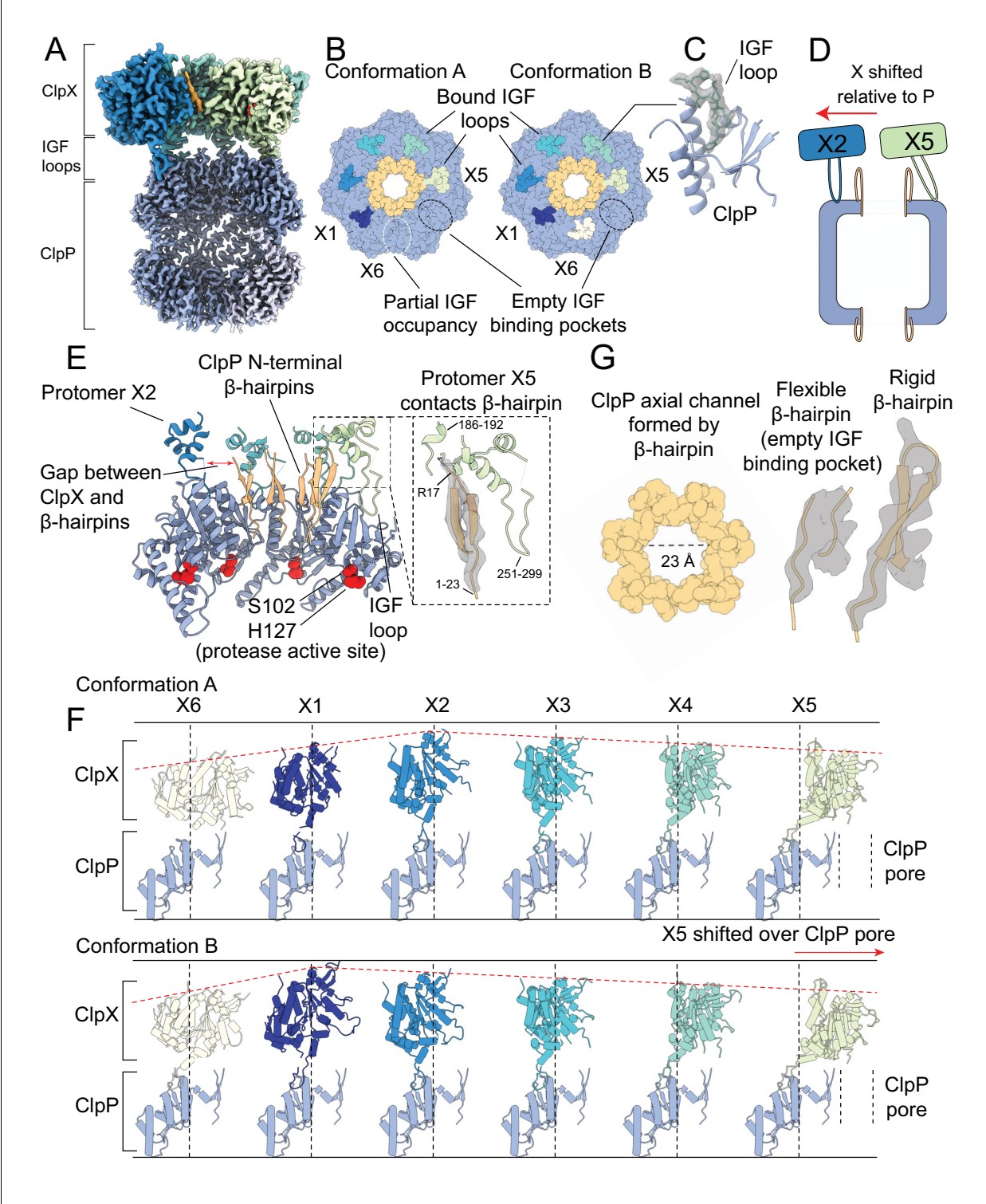

**Figure 2.** The interaction interface between ClpX and ClpP. Protomer X1 occupies the US position in Conformation A and protomer X6 the LS position in Conformation B. (**A**) Cutaway density map of the overall architecture of the ClpXP interaction interface. (**B**) Models for the ClpP apical surface and the ClpX IGF loops (residues 265–275). The empty IGF-binding pocket (dotted oval) resides clockwise to the X5 protomer in both conformations. In Conformation A, an IGF loop was not built into the map for protomer X6 due to weak loop density (yellow dotted line). (**C**) Model in map fit for an IGF loop (grey), surrounded by regions of ClpP comprising the IGF binding site (blue). (**D**) Interaction of the ClpP N-terminal gates with ClpX. The X2 protomer moves away from the gates, while the X5 protomer directly contacts the gate of the ClpP protomer to which its IGF loop is bound, shown in (**E**). Inset shows the details of this interaction, with model in map fit for the ClpP β-hairpin. (**F**) Positions of ClpX protomers relative to ClpP. Images were

*Figure 2 continued on next page*

*Figure 2 continued*

generated by fitting all ClpP protomers to a common protomer and displaying the corresponding ClpX protomer. In both conformations, ClpX adopts a spiral arrangement relative to ClpP (dotted red line). In Conformation A, the X2 protomer is located at the top of the spiral, while in Conformation B the X1 protomer occupies the top position, ~7 Å higher than its position in Conformation A. The ClpX protomers rotate and translate relative to ClpP; protomers X2 and X3 sit nearly atop their ClpP protomers, while X4 and especially X5 show large deviations from this position (vertical lines), with X5 sitting overtop the ClpP axial pore. (G) View down the channel formed by the N-terminal ClpP β-hairpins (left); density and models for the most flexible and rigid β-hairpins (center and right, respectively).

The online version of this article includes the following figure supplement(s) for figure 2:

**Figure supplement 1.** IGF loop flexibility.
**Figure supplement 2.** The ClpP apical loops extend upwards from the ring surface.

activating mechanism of ADEPs involves a disorder-to-order transition of these gates (*Li et al., 2010*), this observation provides direct evidence that ClpX binding has the same effect.

In both Conformations A and B, the rigidification of the ClpP N-terminal gates varies substantially between protomers. At one extreme, the N-terminal residues of the ClpP protomer that is not bound to an IGF loop have the weakest density, appearing to possess the most flexibility at the top of the β-hairpin (*Figure 2G* center; *Figure 2—figure supplement 2E*). In contrast, the N-terminal residues of the ClpP protomer proximal to the X5 ClpX protomer, are held rigidly through contacts with the ATPase (*Figure 2G* right). This interaction is mediated by a pair of α-helices from protomer X5 (residues 254–262 and 291–296) adjacent to the IGF loop, and by another X5 α-helix that precedes the pore-2 loop (residues 186–192). The three α-helices stabilize the β-hairpin structure of the ClpP N-terminal residues by interacting with Arg17 at the top of the hairpin (*Figure 2E* inset). These interactions are unique for the X5 protomer due to the lateral displacement of the ClpX ring relative to the axial pore of ClpP (*Figure 2A*), with the X2 protomer shifted furthest from the pore while the X5 protomer is pulled overtop of the pore (*Figure 2D and F*). Even without interacting directly with ClpX, the remaining five ClpP protomers have nearly rigid N-terminal gates, with reduced density only for residues located in the turn of the β-hairpin that likely reflects increased flexibility in this region (*Figure 2—figure supplement 2F*). Gate rigidification therefore appears to be mediated by an allosteric effect when IGF loops engage their respective ClpP binding pockets, akin to the effect observed upon ADEP binding (*Gersch et al., 2015*; *Li et al., 2010*). In the ClpXP complex with substrate bound, only a single ClpX protomer (X5) interacts directly with a single ClpP gate at any given time. The mechanism of substrate translocation described below involves movement of the ClpX ring on the apical surface of ClpP, with each ClpX protomer forming contacts with a ClpP gate as translocation proceeds.

## Substrate engagement by ClpX

ClpX fits neatly into the emerging consensus for how AAA+ unfoldases engage substrate (*Monroe et al., 2017*; *Yu et al., 2018*; *De la Peña et al., 2018*; *Shin et al., 2019*; *Ripstein et al., 2017*). In both Conformations A and B, five of the six pore-1 loops of ClpX interact tightly with the substrate backbone, with Tyr153 forming the majority of the interface (*Figure 3A and B*). These protomers generate a right-handed spiral that wraps around an 8 to 10 residue stretch of the substrate along the central channel of ClpX (*Figure 3C and D*). In Conformation A, the X1 pore-1 loop in the US orientation is disengaged, with the X2 protomer at the top of the spiral. As the spiral continues downwards, protomers X3, X4, and X5 make contacts with the substrate. The spiral terminates with the X6 pore-1 loop that is engaged with the substrate closest to the ClpP apical surface (*Figure 3C*). In Conformation B, the X6 pore-1 loop is disengaged (LS orientation), while the X1 pore-1 loop engages a new section of substrate (*Figure 3D*) and the X5 protomer is now located closest to the ClpP surface. When transitioning between substrate-engaged and disengaged states, the X6 pore-1 loop undergoes a large conformational change (*Figure 3E*). As it disengages the substrate the 'lasso' fold (*Figure 3E*, left) that allows Tyr153 and Val154 of pore-1 to point toward the substrate is replaced by a short α-helix immediately following the GYVG motif of the loop.

Interestingly, the X1 and X5 protomers make additional contacts with the substrate in Conformation B. Whereas all of the pore-2 loops from the other five protomers are radially removed and disengaged from the substrate, the X1 protomer extends its pore-2 loop residues Ile198 and Thr199 to engage the substrate at a position three or four residues down the substrate chain from where its

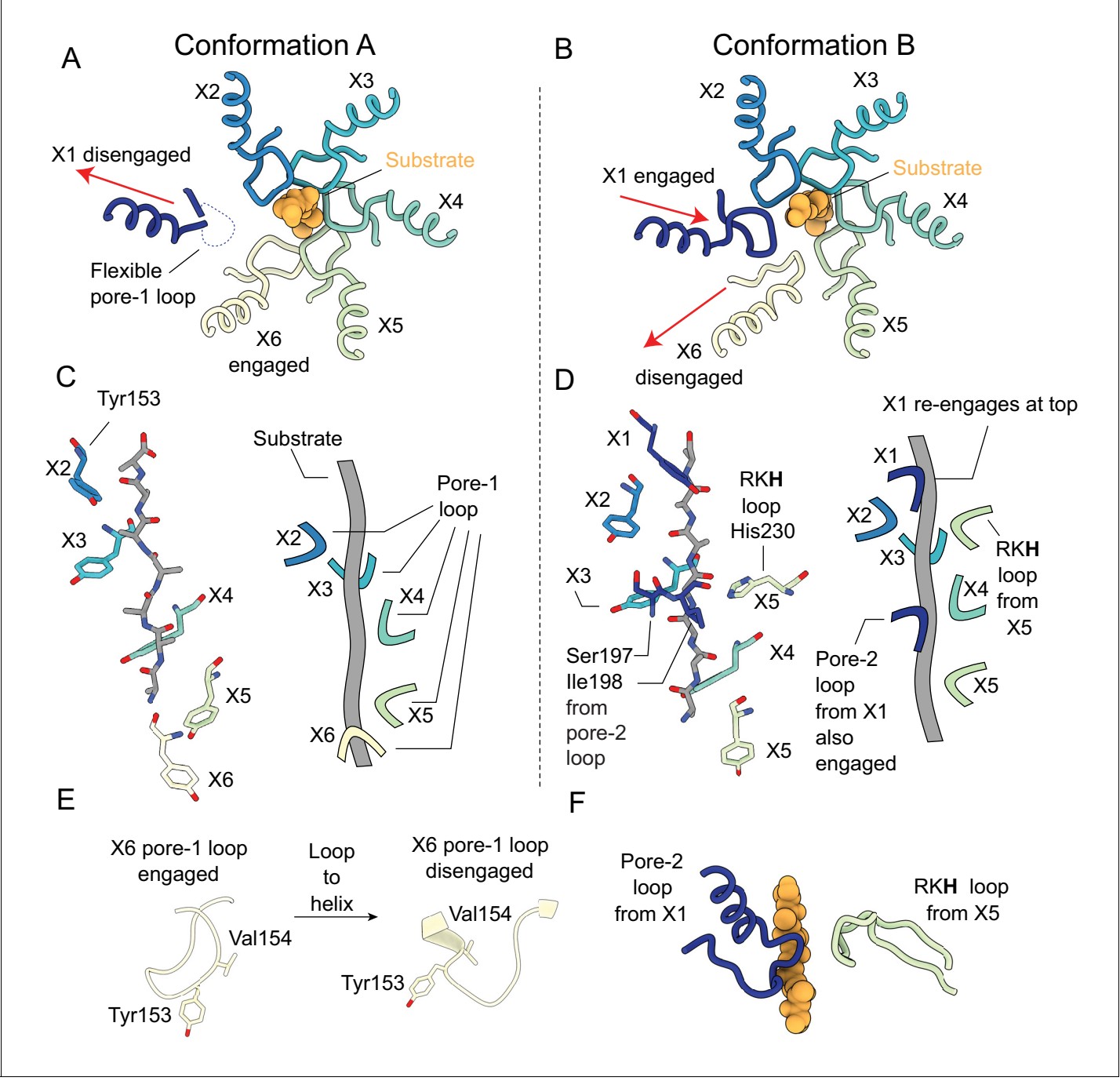

**Figure 3.** Substrate engagement by the ClpX pore loops. (**A and B**) Pore-1 loop residues grip the substrate, as observed in this view looking down the axial channel. The substrate is modeled as poly-Ala. (**A**) In Conformation A, the X1 protomer is disengaged from the substrate in the US position (red arrow) and shows noticeable flexibility in its pore-1 loop. (**B**) In Conformation B, the X1 protomer contacts the substrate but the X6 protomer has disengaged into the LS position (red arrows). (**C and D**) Model of substrate and interacting residues, along with schematic, viewed perpendicular to the axial channel. (**C**) In Conformation A, only the Tyr153 residue of the pore-1 loop makes significant contacts with the substrate; the five protomers (X2, X3, X4, X5, X6) form a downward spiral surrounding the backbone of the substrate chain, extending to the apical surface of ClpP. (**D**) In Conformation B, in addition to the five Tyr153 residues from the five gripping protomers (X1, X2, X3, X4, X5), more contacts are formed: Ser197 and Ile198 from the pore-2 loop of the X1 protomer and His230 from the RKH loop of the X5 protomer. (**E**) Substrate-induced conformational changes of the X6 pore-1 loop upon releasing client. When the X6 protomer is substrate engaged its pore-1 loop forms a lasso like conformation, with Tyr153 and Val154 both pointing toward the axial channel (left). When disengaged, a part of the pore-1 loop becomes an α-helix, with Val154 and Tyr153 no longer pointing outwards (right). (**F**) Architecture of the additional contacts made by the pore-2 and RKH loops from the X1 and X5 protomers, respectively.

*Figure 3 continued on next page*

*Figure 3 continued*

The online version of this article includes the following figure supplement(s) for figure 3:

**Figure supplement 1.** RKH loop positions.

pore-1 loop binds (*Figure 3D and F*). The RKH loops, located at the top of the ClpX ring (*Figure 3—figure supplement 1*), are conserved motifs among the ClpX family of AAA+ unfoldases and are known to contribute to substrate recognition (*Martin et al., 2007*; *Farrell et al., 2007*;

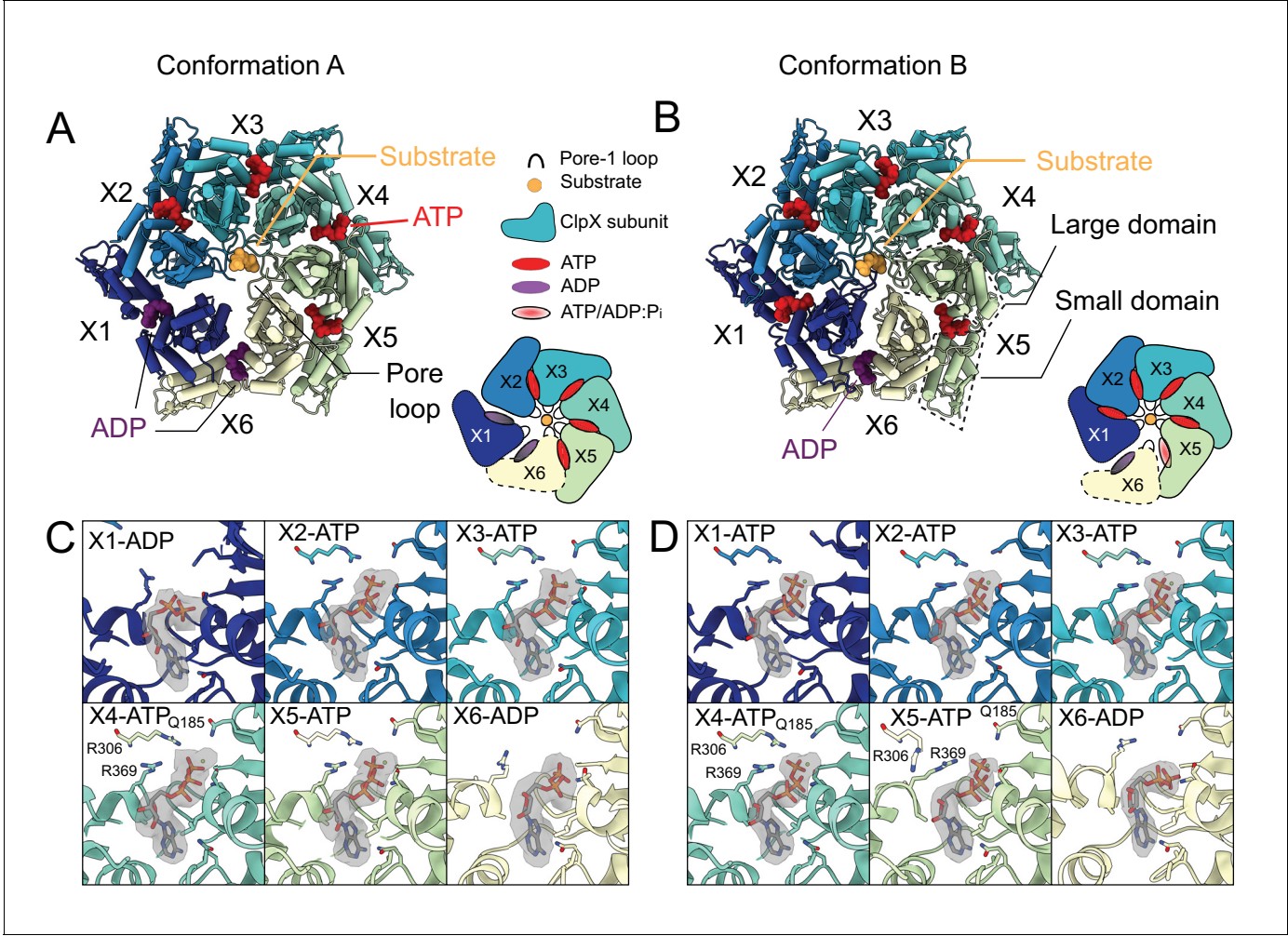

**Figure 4.** Nucleotide occupancy and interactions with ClpX. (A and B) Model of ClpX in conformations A and B, looking into the ClpX pore. Nucleotides are shown and color-coded, with ATP red and ADP purple, and bind between the large and small domains (boxed). Inset shows a schematic of the protomer positions and nucleotide occupancies. Conformation A has two bound ADP, in the X1 (dark blue, US in Conformation A in this representation) and X6 (yellow, LS in Conformation B in this representation) protomers, with the X1 protomer disengaged from the substrate and away from the axial pore (US position). In Conformation B only a single ADP is bound, protomer X1 is reengaged with substrate and X6 has disengaged. (C) Nucleotide-binding sites of Conformation A, with key interacting residues shown along with the experimental density maps corresponding to bound nucleotide. In the ADP-bound sites, both the arginine finger (R306) and the sensor-II arginine (R369) have moved away, while in the ATP bound sites they form close contacts with the β and γ-phosphates. (D) Nucleotide binding in Conformation B. As in Conformation A, R306 and R369 have moved away from the ADP, while in the X5 protomer only the arginine finger has moved away, while the sensor-II transitions closer to the γ-phosphate.

The online version of this article includes the following figure supplement(s) for figure 4:

**Figure supplement 1.** Relative orientation between large and small domains of ClpX.
**Figure supplement 2.** Local resolution maps of ClpX.
**Figure supplement 3.** ATP-binding pocket densities.

**Table 1.** Cryo-EM data acquisition, processing, atomic model statistics, and map/model depositions.

**A. Cryo-EM data acquisition and image processing.**

**Data collection**

| | NmClpXP + GFP-SsrA | apo-NmClpP |
|---|---|---|
| Electron Microscope | Titan Krios | Tecnai F20 |
| Camera | Falcon 3EC | K2 Summit |
| Voltage (kV) | 300 | 200 |
| Nominal Magnification | 75,000 | 25,000 |
| Calibrated physical pixel size (Å) | 1.06 | 1.45 |
| Total exposure (e/Å$^2$) | 42.7 | 35 |
| Exposure rate (e/pixel/s) | 0.8 | 5 |
| Number of frames | 30 | 30 |
| Defocus range (μm) | 0.9 to 1.7 | 0.8 to 3.3 |

**Image processing**

| | | |
|---|---|---|
| Motion correction software | *cryoSPARC v2* | *cryoSPARC v2* |
| CTF estimation software | *cryoSPARC v2* | *cryoSPARC v2* |
| Particle selection software | *cryoSPARC v2* | *cryoSPARC v2* |
| Micrographs used | 2680 | 122 |
| Particle images selected | 466,549 | 100,132 |
| 3D map classification and refinement software | *cryoSPARC v2* | *cryoSPARC v2* |

**B. Map and model statistics.**

| EM maps | NmClpXP with Symmetry | Focused NmClpX Conformation A | Focused NmClpX Conformation B | Apo-NmClpP | |
|---|---|---|---|---|---|
| Particle images contributing to maps | 377,234 | 110,696 | 178,448 | 50,403 | |
| Applied symmetry | D7 | C1 | C1 | D7 | |
| Applied B-factor (Å$^2$) | −116.2 | −84.1 | −88.1 | −229.7 | |
| Global resolution (FSC = 0.143, Å) | 2.3 | 3.3 | 2.9 | 4.1 | |

| Model Building | NmClpX Conformation A focused | NmClpX Conformation B focused | NmClpP with Symmetry | NmClpXP Conformation A combined | NmClpXP Conformation B combined |
|---|---|---|---|---|---|
| Modeling software | Coot, Phenix, Rosetta | | | | |
| Number of residues | 1912 | 2018 | 2671 | 4583 | 4689 |
| RMS bond length (Å) | 0.0043 | 0.0037 | 0.0039 | 0.0044 | 0.0038 |
| RMS bond angle (°) | 1.04 | 0.93 | 0.91 | 1.00 | 0.92 |
| Ramachandaran outliers (%) | 0.00 | 0.05 | 0.00 | 0.00 | 0.02 |
| Ramachandran favored (%) | 98.33 | 97.32 | 95.68 | 97.07 | 96.39 |
| Rotamer outliers | 0.25 | 0.24 | 0.00 | 0.13 | 0.10 |
| C-beta deviations | 0 | 0 | 0 | 0 | 0 |
| Clashscore | 1.45 | 2.8 | 0.67 | 1.84 | 2.11 |
| MolProbity score | 0.88 | 1.20 | 1.02 | 1.11 | 1.22 |
| EMRinger score | 2.78 | 3.42 | 5.6 | | |
| Map-Model CC_mask | 0.75 | 0.78 | 0.82 | | |
| Ligand | 2 ADP 4 Mg-ATP | 1 ADP 5 Mg-ATP | | 2 ADP 4 Mg-ATP | 1 ADP 5 Mg-ATP |

**C. Residues excluded in atomic models*.**

| ClpX protomer | NmClpXP Conformation A combined | NmClpXP Conformation B combined |
|---|---|---|
| X1 (Chain A) | 1–62, 102–109, 149–156, 191–204, 224–236, 263–264, 273–282, 413–414 | 1–62, 228–233, 275–278, 413–414 |
| X2 (Chain B) | 1–62, 102–111, 192–198, 226–235, 275–279, 413–414 | 1–62, 226–233, 275–279, 413–414 |
| X3 (Chain C) | 1–62, 192–198, 225–232, 275–279, 413–414 | 1–62, 192–198, 228–233, 274–278, 413–414 |
| X4 (Chain D) | 1–62, 103–106, 193–198, 226–234, 276–280, 413–414 | 1–62, 193–198, 229–233, 413–414 |
| X5 (Chain E) | 1–62, 193–198, 229–233, 277–281, 413–414 | 1–62, 193–198, 277–279, 413–414 |
| X6 (Chain F) | 1–62, 193–201, 225–234, 259–287, 413–414 | 1–62, 191–204, 227–233, 275–282, 413–414 |

| ClpP subunits | NmClpXP Conformation A combined | NmClpXP Conformation B combined |
|---|---|---|
| Chain H | 1–5, 16–18, 199–204 | 1–5, 16–18, 199–204 |
| Chain I | 1–5, 16–17, 199–204 | 1–5, 16–17, 199–204 |
| Chain J | 1–5, 17, 199–204 | 1–5, 17, 199–204 |
| Chain K | 1–5, 199–204 | 1–5, 199–204 |
| Chain L | 1–5, 16–17, 199–204 | 1–5, 16–17, 199–204 |
| Chain M | 1–5, 13–19, 199–204 | 1–5, 13–19, 199–204 |
| Chain N | 1–5, 16–17, 199–204 | 1–5, 16–17, 199–204 |
| Chain O | 1–5, 16–17, 199–204 | 1–5, 16–17, 199–204 |
| Chain P | 1–5, 16–17, 199–204 | 1–5, 16–17, 199–204 |
| Chain Q | 1–5, 16–17, 199–204 | 1–5, 16–17, 199–204 |
| Chain R | 1–5, 16–17, 199–204 | 1–5, 16–17, 199–204 |
| Chain S | 1–5, 16–17, 199–204 | 1–5, 16–17, 199–204 |
| Chain T | 1–5, 16–17, 199–204 | 1–5, 16–17, 199–204 |
| Chain U | 1–5, 16–17, 199–204 | 1–5, 16–17, 199–204 |

**D. Deposited maps and associated coordinate files.**

| Maps | EMDB code | Associated PDB ID |
|---|---|---|
| NmClpXP Conformation A | EMD-21187 | 6VFS |
| NmClpXP Conformation B | EMD-21194 | 6VFX |
| NmClpXP D7 | EMD-21195 | |
| Apo-NmClpP | EMD-21196 | |

*Protomer X1 is US in Conformation A and protomer X2 is LS in Conformation B.

*Siddiqui et al., 2004*). In Conformation B, only the RKH loop of X5 directly engages substrate, with its His230 residue interacting at a position three or four residues up the substrate chain from where its pore-1 loop binds (*Figure 3D and F*). In order to contact substrate at this position the RKH loop of the X5 protomer rearranges to point toward ClpP (*Figure 3F*), in contrast to the other RKH loops that point away from the axial channel of ClpX (*Figure 3—figure supplement 1*). These five RKH loops are located away from the substrate, and have weaker density in the maps, likely indicating flexibility.

## Nucleotide state and cycling in ClpX

The cryo-EM maps allowed unambiguous identification of the nucleotide bound to each ClpX protomer. In Conformation A, there is strong density for ATP and a magnesium cofactor in protomers X2, X3, X4, and X5, while the X1 and X6 protomers are ADP loaded, with no density for either the γ-phosphate or magnesium (*Figure 4A and C*). As in both the classic and HCLR clades of AAA+ ATPases (*Monroe et al., 2017*; *Yu et al., 2018*; *De la Peña et al., 2018*; *Shin et al., 2019*; *Ripstein et al., 2017*), the orientation between the large and the small AAA+ domains of ClpX changes considerably depending on the nucleotide state. In Conformation A, both the X1 and X6 protomers adopt conformations that allow the X1 protomer to move away from the pore in the US position (*Figure 4—figure supplement 1*) and there is reduced surface area buried between X1 and its neighboring protomers X2 and X6. In agreement with this finding, the local resolution in the cryo-EM maps (*Figure 4—figure supplement 2*) suggests that the X1 protomer in the US position is significantly more flexible than the other protomers. In Conformation B, only the X6 protomer is ADP-bound. The X1, X2, X3, X4, and X5 protomers are ATP-bound with clear density for the γ-phosphate and the magnesium cofactor (*Figure 4B and D*). Interestingly, the X5 protomer adopts a unique orientation between its large domain and small domain compared to the other ATP-bound protomers in order to accommodate the translation of the X6 protomer away from the substrate translocation channel into the LS position (*Figure 4B* orange box; *Figure 4—figure supplement 1*).

In Conformation A, the β- and γ-phosphates of the bound ATP are stabilized by interactions with the sensor-II arginine (Arg369) of the same subunit and the arginine finger (Arg306) from the adjacent clockwise protomer (*Figure 4C*). In the ADP-bound sites of the X1 and X6 protomers, both arginines have moved away, allowing for a more open nucleotide-binding pocket (*Figure 4C*). In Conformation B, both the sensor-II arginine and the arginine finger contact the β and γ-phosphates in only four of the five ATP-bound protomers (X1, X2, X3, and X4) (*Figure 4D*). In the X5 protomer (at the bottom of the spiral), the adjacent X6 protomer and its arginine finger have pivoted away from the nucleotide into the LS position, allowing the sensor-II arginine to move closer to the γ-phosphate of the bound nucleotide. While there is clear density for the γ-phosphate, the resolution of our experimental map was not sufficient to differentiate between ATP and a long-lived post-hydrolysis ADP/P_i state. Notably, this key sensor II-priming motif in the nucleotide site adjacent to an ADP bound site has also been reported for the Lon protease bound to substrate (*Shin et al., 2019*), a close relative in the HCLR clade of AAA+ proteins.

## Discussion

In this study, we used cryo-EM to build atomic models of a ClpXP complex bound to a SsrA-tagged GFP substrate. Structures of a variety of different AAA+ rings with substrates have emerged in the past several years (*Monroe et al., 2017*; *Yu et al., 2018*; *De la Peña et al., 2018*; *Shin et al., 2019*; *Ripstein et al., 2017*), providing a description of how these ATP-dependent molecular machines unfold substrates. In addition, methyl-TROSY NMR studies of a VAT-substrate complex provided atomic-level insights into structural changes associated with the substrate as it is pulled into the lumen of the unfoldase (*Augustyniak and Kay, 2018*). Nevertheless, a detailed understanding of how a pair of symmetry mismatched ClpP/ClpX rings can work in tandem to unfold, translocate, and subsequently degrade substrate in a processive manner has remained elusive. Our structural data suggest a translocation mechanism that can be explained in terms of a pair of conformers, denoted as Conformations A and B here, that represent two steps along the substrate translocation pathway (*Figure 5A*). To aid in the following discussion, we have color-coded the ClpX protomers and refer to them with their X1 - X6 labels. As we will discuss, each of these protomers will cycle between all six protomer positions along the spiral, including, among others, a state primed to hydrolyze ATP (denoted as ATP*) as well as the LS and US positions. This cycling is linked to ATP hydrolysis, to the attachment position of ClpX pore-loops along the substrate, and to the interaction of ClpX IGF loops with ClpP. We begin our discussion of the unfolding/translocation cycle with the X1 protomer in Conformation A in the US position, detached from substrate and bridging the lowest and highest positions of the ClpX spiral. Both X1 and X6 protomers are ADP-bound, while the remaining protomers are in ATP-bound states. In Conformation B, X1 moves to the highest position of the spiral with respect to substrate (compare *Figure 2F*, top and bottom panels) simultaneously exchanging ADP for ATP in its nucleotide-binding pocket (*Figure 5B*). This transition brings X1 into close

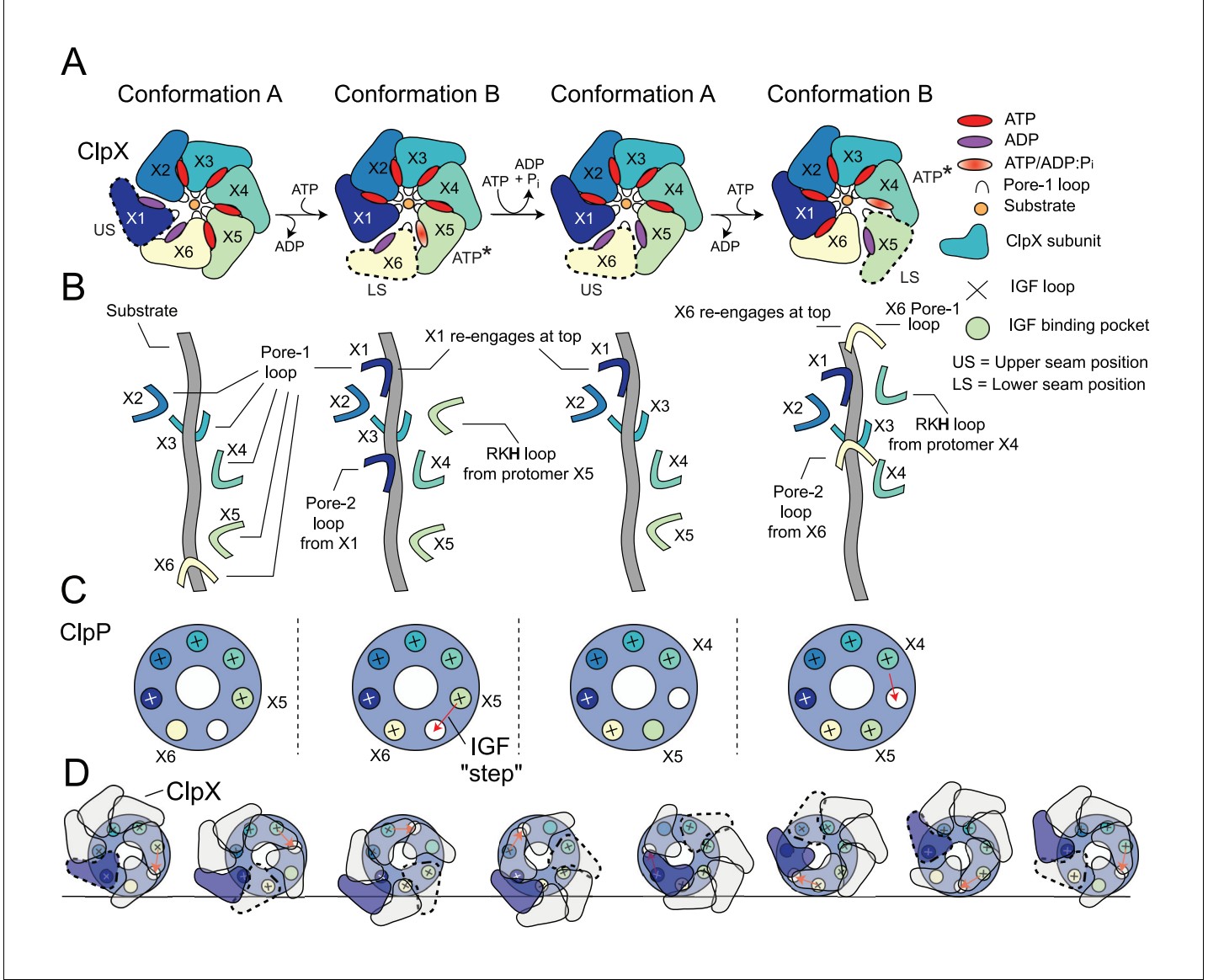

**Figure 5.** Translocation model for ClpXP. (**A**) Schematic of the ClpX-nucleotide-dependent conformational cycle. The transition from Conformation A to Conformation B is mediated through nucleotide exchange in the US position (1st to 2nd ring), with ATP hydrolysis and phosphate release in the ATP* position restoring the A conformation from the B state (2nd to 3rd ring) so that the cycle can repeat. In the new Conformation A (3rd ring), protomer X6 (yellow) adopts the US conformation and the X1 protomer (previous US) moves to the top of the spiral. In this way each protomer of the complex cycles through all positions as ATP is hydrolyzed in successively new ATP* protomer states, requiring the hydrolysis of seven ATPs. (**B**) Hand-over-hand mechanism of pore-loop mediated translocation. In the first A to B transition depicted in (A) X1 (US protomer) reengages at the top of the spiral and the LS pore one loop (X6 in Conformation B) disengages from the bottom of the spiral. As this is repeated the pore loops move 'up' the substrate and pull it into the degradation chamber. (**C**) IGF loop movements accompanying the hand-over-hand movements of ClpX. Shown in the small circles are the seven IGF-loop-binding sites in ClpP with a '+' sign indicating engagement with an IGF loop of a given ClpX protomer. The binding sites are color coded according to the color of the protomer that interacts with the corresponding site. In conformation A, the IGF loop of X6 is unbound or only weakly bound, while in Conformation B it is bound. A new Conformation A is generated when the IGF loop from the ATP* protomer unbinds. It subsequently reengages with ClpP in a new Conformation B (third and fourth rings in (A)). In this way, the IGF loops 'step' around the ClpP ring, with each complete rotation requiring the hydrolysis of seven ATPs. (**D**) Schematic of the ClpX-ClpP interactions during seven ATP hydrolysis steps that lead to a net rotation of ClpX protomers by 60°. Because of the six- to seven-fold symmetry mismatch a total of forty-two ATPs must be hydrolyzed to return the protomers back to their original starting positions in the cycle (i.e. ring 1). Each of the seven ring structures corresponds to Conformation A, with the red arrows indicating the subsequent step of the IGF loop of the ATP* protomer along the cycle. The US protomer in each step is highlighted by the dashed trace.

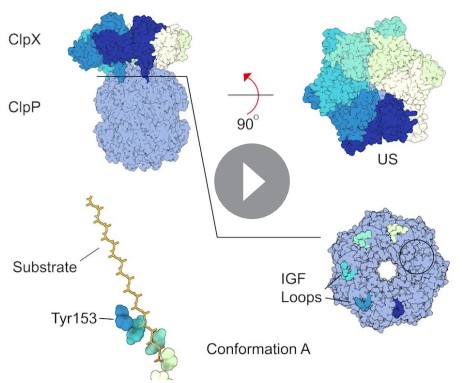

**Video 2.** Hand-over-hand translocation cycle of the ClpXP complex. Interpolation between states A → B → A → B. Side and top views of the complex are shown (top), along with the pore-1 loop Tyr153 and substrate (bottom left) as well as IGF loops (bottom right). https://elifesciences.org/articles/52158#video2

proximity with X2 and allows engagement of X1's pore loops with substrate. The X1 pore-I loop now sits at the top of the spiral, two substrate residues above the X2 pore loop (*Figure 5A and B*), thereby translocating additional residues of substrate into the unfoldase. This transition also leads to release of substrate by X6, which pivots away from the central pore to adopt the LS position (second ring, *Figure 5A*). ATP hydrolysis and phosphate release by X5, which was in the ATP-hydrolysis primed ATP* state, restores the ClpX spiral back to the Conformation A state, (third ring of the four in *Figure 5A*), with X6 now assuming the US position. The complex is thus reset so that it can take another step along the substrate during the next A to B transition (third ring to fourth ring in *Figure 5A*). The cumulative effect of these transitions is a cycling of ClpX protomers through different positions on the spiral (*Figure 5A and B*). For example, the protomer initially in the US position (X1 in our discussion) will, with each step, move along the spiral such that it transitions from the US position to the top of the spiral (Conformation B) and then successively moving to lower spiral positions, through the ATP* state and finally the LS position, as substrate is translocated. Each Conformation A→Conformation B→Conformation A step results in the exchange of ADP for ATP and a single ATP hydrolysis event. Continuous repetition of these steps leads to a processive 'hand-over-hand' translocation of substrate through the axial pore of ClpX into ClpP (Illustrated in *Video 2*).

The sequential hydrolysis of ATP at the ATP* position results in unidirectional substrate translocation and movement of pore loops along the substrate (*Figure 5B*), with the counter clockwise hydrolysis cycle translating into a repeated downward pulling force on the substrate. Both the sequential hydrolysis of ATP as well as the 'hand-over-hand' substrate pulling model appear to be conserved for AAA+ unfoldases (*Monroe et al., 2017*; *Yu et al., 2018*; *De la Peña et al., 2018*; *Shin et al., 2019*; *Ripstein et al., 2017*).

The cycle described above, in which each protomer eventually adopts all positions in the spiral, has implications for how the IGF loops interact with the ClpP binding pockets. If the spiral position of each ClpX protomer was independent of IGF binding, alignment of the asymmetric ClpX spiral would lead to averaging of the IGF loops into all the pockets of ClpP during cryo-EM map refinement. However, our structures clearly show an empty IGF pocket on ClpP *only* between the ATP* and LS positions of Conformation B (*Figure 2B*), suggesting that this pocket must undergo a 'cycle' as well (*Figure 5C*). Thus, following ATP hydrolysis and phosphate release, the IGF loop of the protomer in the ATP* position (X5 initially) must leave its engaged pocket and move to bind the adjacent empty site on ClpP (red arrows in *Figure 5C*), as the complex transitions back to Conformation A (third ring in *Figure 5A*). In this state, the IGF loop appears not to have assumed a structure

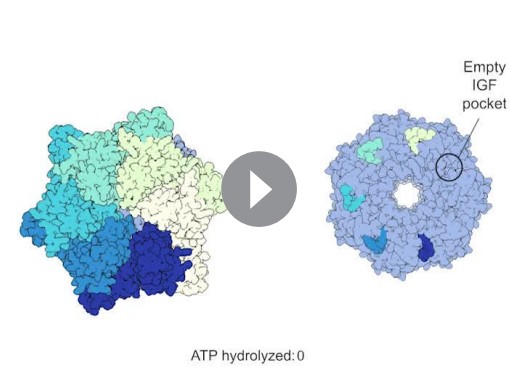

**Video 3.** ClpX precession on the ClpP apical surface. Repeated interpolation between states reveals a precession of the ClpX ring as its IGF loops are placed successively in different ClpP binding sites. The position of the empty binding pocket on ClpP undergoes a complete cycle around the complex for every 60° rotation of the ClpX ring relative to the ClpP ring. https://elifesciences.org/articles/52158#video3

that allows for tight binding as only weak density is observed for it in EM maps. In contrast, strong binding is observed for this loop in Conformation B (*Figure 4—figure supplement 1*). Consistent with this model, the IGF loop of the X5 protomer has an 'extended/stretched' structure in the ATP* position of Conformation B (*Figure 2F*), which likely causes strain and facilitates loop release and subsequent binding to the corresponding site on the adjacent ClpP protomer (*Figure 5C*). After seven IGF loop release and reengagement events, accompanied by seven ATP hydrolysis steps, the 'empty' ClpP binding site makes a complete cycle, returning to its starting position (*Figure 5C*). The accompanying motion of ClpX on the apical surface of ClpP for a given IGF loop transition is complex and subtle, yet seven of these transitions lead to a net 60° rotation of ClpX with respect to ClpP (note the positions of the purple ClpX protomer in the first and seventh panels in *Figure 5D*; *Video 3*). The model described above is the simplest one that is consistent with our data. While our structures suggest that a sequential hydrolysis model exists for ClpXP, more complicated models can be envisioned that require additional states for which we have no experimental data. Indeed biochemical evidence suggests the possibility that when one or more subunits are catalytically deficient a more complex mechanism may be operational, allowing ClpXP to deviate from a strictly sequential hydrolysis scheme (*Martin et al., 2008b*).

During preparation of this manuscript, two manuscript preprints reported structures of ClpXP from *L. monocytogenes* (*Gatsogiannis et al., 2019*) and from *E. coli* (*Fei et al., 2019*), with ClpX resolutions ranging from 3.9 to 6 Å. While density maps and atomic models are not available for direct comparison with the NmClpXP, a few important similarities and differences are notable. In all species, the ClpX ring appears to adopt a similar offset relative to the symmetry axis of ClpP, indicating that the general architecture of the ClpXP machinery is conserved. Notably, however, the N-terminal apical loops of *L. monocytogenes* ClpP are not rigidified upon ClpX binding (*Gatsogiannis et al., 2019*), in contrast to the observation here that the gates rigidify upon binding to create a pore for substrate entry. In the case of *L. monocytogenes* ClpXP, unusual head-to-head dimers were observed that appeared to be mediated by the zinc binding domains. Despite the presence of the zinc-binding domains in our *N. meningitidis* construct, no such dimers were observed. While this difference may be due to differences across species, we note that the glutaraldehyde chemical cross-linker used to stabilize the *L. monocytogenes* ClpXP complex may have induced the formation of artefactual dimers.

Notably, substrate engagement is different between the ClpXP from *N. meningitidis* described here and from *E. coli* (*Fei et al., 2019*). In Conformation A seen with *N. meningitidis*, there is no engagement of substrate by any of the six pore-2 or six RKH loops, while in Conformation B only a single RKH loop and a single pore-2 loop contact substrate (*Figure 3D and F*). This observation is in contrast to the recently reported *E. coli* ClpXP structures (*Fei et al., 2019*) in which arginines located in the pore-2 loop from three to five different protomers (depending on the structure), as well as multiple RKH loops, form major contacts with substrate. The differences in these interactions may reflect the fact that while pore-1 loop and RKH loop motifs are critical for forming interactions with substrate, and in the case of pore-1 loops in particular, in providing the force for translocation, additional contacts may differ between species and may be necessary for recognition of degradation signals and for substrate specificity. The RKH loops are essential for the recognition of SsrA-tagged substrates by *E. coli* ClpX, while human ClpX, which substitutes the RKH sequence with an RKL tripeptide and contains pore-1 loops with the same sequence as *E. coli* ClpX, fails to recognize substrates with the SsrA-tag. Notably, however, an engineered human construct bearing the pore-2 and RKH loops of *E. coli* ClpX is able to unfold SsrA-tagged substrates (*Farrell et al., 2007*). Reordering of RKH loop residues to KHR and KRH results in significantly impaired substrate degradation (*Fei et al., 2019*) and a more disrupting AAA mutant does not support ClpP degradation of a SsrA-tagged substrate (*Fei et al., 2019*). Furthermore, the single point mutation RKH to RKA results in loss of activity for both SsrA and λO tagged substrates. The tight contact between H230 and the substrate in our maps provides a structural basis for understanding these mutagenesis results.

Although in the ClpXP structures from *E. coli* (*Fei et al., 2019*) the ClpX ring adopts the same spiral staircase arrangement observed in many AAA+ ATPases (*Monroe et al., 2017*; *Yu et al., 2018*; *De la Peña et al., 2018*; *Shin et al., 2019*; *Ripstein et al., 2017*; *Enemark and Joshua-Tor, 2006*; *Huang et al., 2016*), strongly supporting the 'hand-over-hand' mechanism of substrate translocation, the authors of this work instead posit a stochastic mechanism (*Fei et al., 2019*; *Martin et al., 2005*) for the HCLR clade of ATPases based on the observation of only a single bound ADP in their

structures. In the stochastic model, only a single protomer pulls the substrate, via a state that has yet to be observed. Such a model, involving stochastic nucleotide hydrolysis, appears inconsistent with the observations here, with those for the related Lon protease (*Shin et al., 2019*) where two neighboring protomers are bound with ADP, and with a recent structural model for a substrate-bound *E. coli* ClpAP complex showing that the unbinding and rebinding of flexible IGL loops is coupled to substrate translocation as each protomer makes its way from the bottom to the top of the spiral (*Lopez et al., 2019*). The proposed mechanism in that case is similar to the one suggested here for *N. meningitides* ClpXP. The identification of two conformations in the *N. meningitidis* enzyme and observations of the bound nucleotides in each ClpX protomer leads to a simple model of cyclic hydrolysis that, in turn, results in unidirectional substrate translocation. In this model, the substrate is always gripped by multiple protomers, with only a single protomer disengaged as it traverses from the bottom to the top of the spiral. While unique features will undoubtedly continue to emerge regarding subtle functional aspects of different AAA+ unfoldases it is likely that the translocating forces generated by these molecular machines are based on a common mechanism.

# Materials and methods

## Key resources table

| Reagent type (species) or resource | Designation | Source or reference | Identifiers | Additional information |
|---|---|---|---|---|
| Gene (*Neisseria meningitidis*) | clpX | | UniProtKB - Q9JYY3 | |
| Gene (*Neisseria meningitidis*) | clpP | | UniProtKB - Q9JZ38 | |
| Strain, strain background (*Escherichia coli*) | BL21(DE3) | Sigma-Aldrich | CMC0016 | Chemically competent cells |
| Recombinant DNA reagent | pET28-NmClpP | Synthetic (GenScript) | UniProtKB - Q9JYY3 | Plasmid containing ClpP |
| Recombinant DNA reagent | pET28a-NmClpX | Synthetic (GenScript) | UniProtKB - Q9JZ38 | Plasmid containing ClpX |
| Recombinant DNA reagent | pET28a-GFP-SsrA | (*Ripstein et al., 2017*) | | Plasmid containing GFP-SsrA |
| Chemical compound, drug | PKM-AMC | GenScript | | Fluorogenic peptide for protease assays |
| Software, algorithm | *EPU* | Thermo Fischer Scientific | | EM imaging software |
| Software, algorithm | *cryoSPARC* v2 | (*Punjani et al., 2017*) | RRID: SCR_016501 | EM reconstruction software |
| Software, algorithm | UCSF Chimera | (*Pettersen et al., 2004*) | RRID: SCR_004097 | Molecular Visualization Software |
| Software, algorithm | UCSF ChimeraX | (*Goddard et al., 2018*) | RRID: SCR_015872 | Molecular Visualization Software |
| Software, algorithm | *Coot* | (*Emsley and Cowtan, 2004*) | RRID: SCR_014222 | Protein Model Building Software |
| Software, algorithm | Phyre2 | (*Kelley et al., 2015*) | RRID: SCR_010270 | Protein Model Building Software |
| Software, algorithm | Rosetta | (*Wang et al., 2015*) | RRID: SCR_015701 | Protein Model Building Software |
| Software, algorithm | Phenix | (*Adams et al., 2010*) | RRID: SCR_014224 | Protein Model Building Software |
| Software, algorithm | Molprobity | (*Arendall III et al., 2010*) | RRID: SCR_014226 | Protein Model Evaluation Software |
| Software, algorithm | EMRinger | (*Barad et al., 2015*) | | Protein Model Evaluation Software |

## Plasmids and constructs

Codon-optimized genes encoding NmClpX (Uniprot entry: Q9JYY3) bearing an N-terminal His$_6$-TEV affinity tag and ClpP (Uniprot entry: Q9JZ38) with an N-terminal His$_6$-SUMO tag were synthesized by GenScript (Piscataway, NJ) and cloned into the NdeI and BamHI sites of pET28a+ (Novagen, Madison, WI). Point mutations were introduced with the Quikchange mutagenesis method (Agilent, Santa Clara, CA).

## Expression and purification of NmClpP and NmClpX

Transformed Codon+ *E. coli* BL21(DE3) cells were grown in LB media at 37°C. Protein over-expression was induced by addition of 0.2 mM IPTG at OD$_{600}$ = 1.0 and was allowed to proceed overnight at 18°C. Cells were lysed in buffer containing 50 mM Tris, 300 KCl, 10 mM imidazole, 10% glycerol, pH 7.0 and NmClpP and NmClpX proteins purified by Ni-affinity chromatography [HisTrap HP (GE)] in lysis buffer. Bound proteins were eluted from the Ni column by increasing the imidazole concentration to 500 mM. The affinity tag was removed by the addition of TEV (for ClpX) or Ulp1 (for ClpP) protease followed by dialysis against lysis/wash buffer that included 5 mM DTT. Following a reverse Ni-affinity chromatography step, the flow-through, free from the cleaved tag and other impurities, was concentrated with an Amicon Ultra-15 50K MWCO (Millipore) concentrator and subjected to size exclusion chromatography (SEC) with a Superdex 200 Increase 10/300 (GE) column in SEC buffer (50 mM imidazole, 100 mM KCl, 5 mM DTT, pH 7.0). Fractions corresponding to NmClpX and NmClpP were pooled and stored at 4°C in the SEC buffer until further use. Salts containing magnesium were avoided during the purification of NmClpX to prevent protein aggregation. Protein concentrations were determined in 8 M GdnCl using extinction coefficient values (7450 M$^{-1}$ cm$^{-1}$ for ClpP, 8940 M$^{-1}$ cm$^{-1}$ for ClpX) determined with ExPASy's ProtParam (*Gasteiger et al., 2005*).

## Expression and purification of GFP-SsrA

Green fluorescent protein (GFP) bearing an 11-residue SsrA degradation tag at its carboxyl terminus and a non-cleavable N-terminal His × 6 tag was purified by Ni affinity chromatography followed by SEC on a HiLoad 16/60 Superdex 75 pg column (GE).

## Peptidase rate measurements of NmClpP as a function of pH

The peptidase activity of NmClpP was measured at 37°C with Acetyl-L-Pro-L-Lys-L-Met bearing a C-terminal fluorogenic 7-amino-4-methylcoumarin group (PKM-AMC) as substrate. The reaction was followed with a Synergy Neo2 96-well microplate reader making a measurement every 21 s for 60 min at $\lambda_{ex}$: 355 nm, $\lambda_{em}$: 460 nm. Each well contained 1 μM NmClpP (monomer concentration), 250 μM PKM-AMC, 50 mM citrate, 50 mM phosphate, 50 mM Tris, 100 mM KCl in a total volume of 100 μL adjusted to the appropriate pH. Activities are derived from initial rates extracted and analyzed using a python script written in-house. Standard errors are calculated from repeating each reaction in triplicate.

## GFP-SsrA degradation assays

Degradation of 1 μM samples of GFP-SsrA was followed by the loss of GFP fluorescence ($\lambda_{ex}$: 480 nm, $\lambda_{em}$: 508 nm) with a Synergy Neo2 96-well microplate reader at 25°C. The wells included an ATP-regeneration system (*Nørby, 1988*) that contained 1.5 mM phosphoenolpyruvate, 0.2 mM NADH, 40 μg/mL pyruvate kinase, 40 μg/mL lactate dehydrogenase, and 2 mM MgATP at pH 8.2. In some assays, solutions also contained WT NmClpX at 0.5 μM (hexamer) and/or WT ClpP at a 0.25 μM (tetradecamer), as indicated in *Figure 1B*. All assays were performed in triplicate.

## Preparation of samples for cryo-EM

A 1 mL mixture containing 10 μM (tetradecamer) NmClpP and 20 μM NmClpX together with 200 μM GFP-SsrA was incubated with 20 mM MgATP for 10 min at room temperature. This mixture was applied to a Superdex 200 Increase 10/300 (GE) column equilibrated with 50 mM bicine, 100 mM KCl, 2 mM MgATP, pH adjusted to 8.2 at room temperature (equivalent to pH 8.5 at 4°C – see *Appendix 1—figure 1*), as the running buffer. Following SEC, a 0.5 mL fraction (denoted with a * in *Figure 1—figure supplement 1*) containing doubly capped ClpXP bound to GFP-SsrA was

supplemented with 20 µM GFP-SsrA and vitrified immediately without the addition of any cross-linking agent or detergent.

## Sample vitrification

2.5 µL of the sample mixtures were applied to nanofabricated holey gold grids (*Marr et al., 2014*; *Russo and Passmore, 2014*; *Meyerson et al., 2015*) with a hole size of ~1 µm, that had been glow discharged in air for 15 s. Grids were blotted on both sides using a FEI Vitrobot mark III for 15 s at 4°C and ~100% relative humidity before freezing in a liquid ethane/propane mixture (*Tivol et al., 2008*).

## Electron microscopy

NmClpXP was imaged with a Thermo Fisher Scientific Titan Krios G3 microscope operating at 300 kV and equipped with a FEI Falcon III DDD camera. Structures were calculated from counting mode movies consisting of 30 frames, obtained over a 60 s exposure with defocuses ranging from 0.9 to 1.7 µm. Movies were at a nominal magnification of 75,000 × corresponding to a calibrated pixel size of 1.06 Å and with an exposure of 0.8 electrons/pixel/s, giving a total exposure of 43 electrons/Å$^2$. 2680 movies were collected using the microscope's *EPU* software. The Apo-NmClpP structure (with no ClpX or GFP present) was calculated from data obtained using a FEI Tecnai F20 electron microscope operating at 200 kV and equipped with a Gatan K2 Summit direct detector device camera. Movies consisting of 30 frames over a 15 s exposure were obtained with defocuses ranging from 1.7 to 2.9 µm. Movies were collected in counting mode at a nominal magnification of 25,000 × corresponding to a calibrated pixel size of 1.45 Å and with an exposure of 5 electrons/pixel/s, and a total exposure of 35 electrons/Å$^2$. 122 movies were collected using Digital Micrograph software.

## EM image analysis

Patch based whole frame alignment and exposure weighting was performed in *cryoSPARC* v2 (*Punjani et al., 2017*) with a 10 × 10 grid and the resulting averages of frames were used for patch based contrast transfer function (CTF) determination. Templates for particle selection were generated by 2D classification of manually selected particles. Particle images were extracted in 300 × 300 pixel boxes for further analysis. *Ab inito* map calculation was performed on a random subset of 30,000 particle images, generating an initial map showing density for the complex of a ClpP tetradecamer bound to two ClpX hexamers. A single round of 2D classification was used to remove images of damaged particles and other contaminants from a dataset of 466,549 particle images, with selected classes leaving 377,234 particle images for further analysis. Homogeneous refinement of these particle images using D7 symmetry yielded a map of the complex, with good density for only the ClpP portion, at 2.3 Å resolution.

To improve the density of the ClpX portion of the map, a round of *Ab initio* classification was performed using three classes, of which two classes containing 289,144 particle images had good density for ClpX, with the remaining class containing mostly density for ClpP. Refinement of these particle images with C1 symmetry resulted in a map of the ClpXP complex at a nominal resolution of 2.8 Å, but with poorly defined density for much of ClpX. To improve the interpretability of the map in the ClpX region, local refinement was performed with a mask around the six ClpX subunits without performing signal subtraction for ClpP. This refinement greatly improved the map in the ClpX region, while blurring the density at the distal ClpP ring (indicative of flexibility between ClpX and ClpP). However, density for two of the ClpX subunits remained fragmented and at lower resolution. To help resolve the heterogeneity of this region '3D variability analysis' was performed, which utilizes principle component analysis to separate conformations. Clustering was performed along three eigenvectors. Two clusters were identified along a single eigenvector corresponding to two conformations of ClpX bound to substrate. From the trajectory identified, the two endpoints were used to seed a heterogeneous classification in which the O-EM learning rate was reduced 10-fold to preserve the original character of the seeds yielding two classes split ~40%:60% with 110,696 and 178,448 particle images for Conformations A and B respectively. Subsequent non-uniform refinement yielded maps at 3.3 Å and 2.9 Å respectively, which were then used for model building.

For Apo-NmClpP the same preprocessing steps were applied as described above, and 100,132 particle images were extracted in 160 × 160 pixel boxes for further analysis. Rounds of 2D classification and *ab initio* classification led to a subset of 50,400 particle images that were used to refined a map to 4.1 Å resolution.

### Atomic model building and refinement

To model NmClpP, the crystal structure of NmClpP (PDBID: 5DKP) (*Goodreid et al., 2016*) was rigidly fit with UCSF Chimera (*Pettersen et al., 2004*) into the 2.3 Å D7 symmetric map, followed by relaxation with Rosetta using the density map as an additional term in the scoring function (*Wang et al., 2015*), and utilizing D7 non crystallographic symmetry. The best scoring model was then rigidly fit into the C1 symmetry focussed maps for Conformations A and B. Visual inspection and real space refinement in *Coot* (*Emsley and Cowtan, 2004*) was then used to better fit the model into the density where it deviated from the ideal symmetry version, specifically in areas where ClpX contacted ClpP, and the apical loops (N-terminal β-hairpins).

For NmClpX, Phyre2 (*Kelley et al., 2015*) was used to perform one-to-one threading onto the previous crystal structure of ClpX from *E. coli* (PDBID 3HWS chain A) (*Glynn et al., 2009*). A single chain was then rigidly docked into the X3 position of the 2.9 Å map, and real space refinement and *Ab initio* model building of regions that poorly fit the density, as well as for regions missing from the homology model was performed in *Coot* (*Emsley and Cowtan, 2004*). This model was then relaxed with Rosetta (*Wang et al., 2015*), and rigidly fit into the density for the other five ClpX protomers. Iterative rounds of real space refinement and *Ab initio* model building in *Coot* (*Emsley and Cowtan, 2004*), relaxation in Rosetta (*Wang et al., 2015*), and real space refinement in Phenix (*Adams et al., 2010*), were then used to create the final model. To model Conformation A, the model for Conformation B was used as a starting point before iterative rounds of real space refinement and Ab initio model building in *Coot*, relaxation in Rosetta, and real space refinement in Phenix. While the experimental density for the substrate showed some bulky side chains, attempts to register the SsrA sequence in the density were unsuccessful and the substrate was modeled as polyalanine. Models were evaluated with Molprobity (*Arendall III et al., 2010*) and EMRinger (*Barad et al., 2015*; *Table 1*). Figures and movies were generated in UCSF Chimera (*Pettersen et al., 2004*) and UCSF ChimeraX (*Goddard et al., 2018*), and colors chosen with ColorBrewer (*Harrower and Brewer, 2003*).

## Acknowledgements

We thank Dr. Samir Benlekbir for cryo-EM data collection on the Titan Krios. The authors are grateful to Professor Andreas Martin for helpful discussions. ZAR and SV were supported by a scholarship and a postdoctoral fellowship from the Canadian Institutes of Health Research, respectively. LEK and JLR were supported by the Canada Research Chairs program. This research was funded by Canadian Institutes of Health Research grants FDN-503573 (LEK), PJT-162186 (JLR), and PJT-148564 (WAH). Titan Krios cryo-EM data were collected at the Toronto High-Resolution High-Throughput cryo-EM facility supported by the Canadian Foundation for Innovation and Ontario Research Fund.

## Additional information

#### Competing interests

Lewis E Kay: Reviewing editor, *eLife*. The other authors declare that no competing interests exist.

#### Funding

| Funder | Grant reference number | Author |
|---|---|---|
| Canadian Institutes of Health Research | FDN-503573 | Lewis E Kay |
| Canadian Institutes of Health Research | PJT-162186 | John L Rubinstein |

| Canadian Institutes of Health Research | PJT-148564 | Walid A Houry |
|---|---|---|
| Canadian Institutes of Health Research | | Zev A Ripstein Siavash Vahidi |
| Canada Research Chairs | | Lewis E Kay John L Rubinstein |

The funders had no role in study design, data collection and interpretation, or the decision to submit the work for publication.

### Author contributions

Zev A Ripstein, Conceptualization, Resources, Data curation, Formal analysis, Supervision, Funding acquisition, Validation, Investigation, Visualization, Methodology; Siavash Vahidi, John L Rubinstein, Conceptualization, Resources, Data curation, Formal analysis, Supervision, Funding acquisition, Investigation, Visualization, Methodology; Walid A Houry, Conceptualization, Writing - review and editing; Lewis E Kay, Conceptualization, Resources, Supervision, Funding acquisition, Investigation, Project administration

### Author ORCIDs

Zev A Ripstein https://orcid.org/0000-0003-3601-0596
Siavash Vahidi https://orcid.org/0000-0001-8637-3710
John L Rubinstein https://orcid.org/0000-0003-0566-2209
Lewis E Kay https://orcid.org/0000-0002-4054-4083

### Decision letter and Author response

Decision letter https://doi.org/10.7554/eLife.52158.sa1
Author response https://doi.org/10.7554/eLife.52158.sa2

## Additional files

### Supplementary files

• Transparent reporting form

### Data availability

CryoEM maps and models have been deposited in the EMDB and PDB.

The following datasets were generated:

| Author(s) | Year | Dataset title | Dataset URL | Database and Identifier |
|---|---|---|---|---|
| Ripstein ZA, Vahidi S, Houry WA, Rubinstein JL, Kay LE | 2020 | Maps NmClpXP Conformation A | http://www.rcsb.org/structure/6vfs | RCSB Protein Data Bank, 6VFS |
| Ripstein ZA, Vahidi S, Houry WA, Rubinstein JL, Kay LE | 2020 | Maps NmClpXP Conformation B | http://www.rcsb.org/structure/6vfx | RCSB Protein Data Bank, 6VFX |
| Ripstein ZA, Vahidi S, Houry WA, Rubinstein JL, Kay LE | 2020 | Maps NmClpXP Conformation A | https://www.ebi.ac.uk/pdbe/entry/emdb/EMD-21187 | Electron Microscopy Data Bank, EMD-21187 |
| Ripstein ZA, Vahidi S, Houry WA, Rubinstein JL, Kay LE | 2020 | Maps NmClpXP Conformation B | https://www.ebi.ac.uk/pdbe/entry/emdb/EMD-21194 | Electron Microscopy Data Bank, EMD-21194 |
| Ripstein ZA, Vahidi S, Houry WA, Rubinstein JL, Kay LE | 2020 | Maps NmClpXP D7 | https://www.ebi.ac.uk/pdbe/entry/emdb/EMD-21195 | Electron Microscopy Data Bank, EMD-21195 |
| Ripstein ZA, Vahidi S, Houry WA, Rubinstein JL, Kay LE | 2020 | Maps Apo-NmClpP | https://www.ebi.ac.uk/pdbe/entry/emdb/EMD-21196 | Electron Microscopy Data Bank, EMD-21196 |

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

## Appendix 1

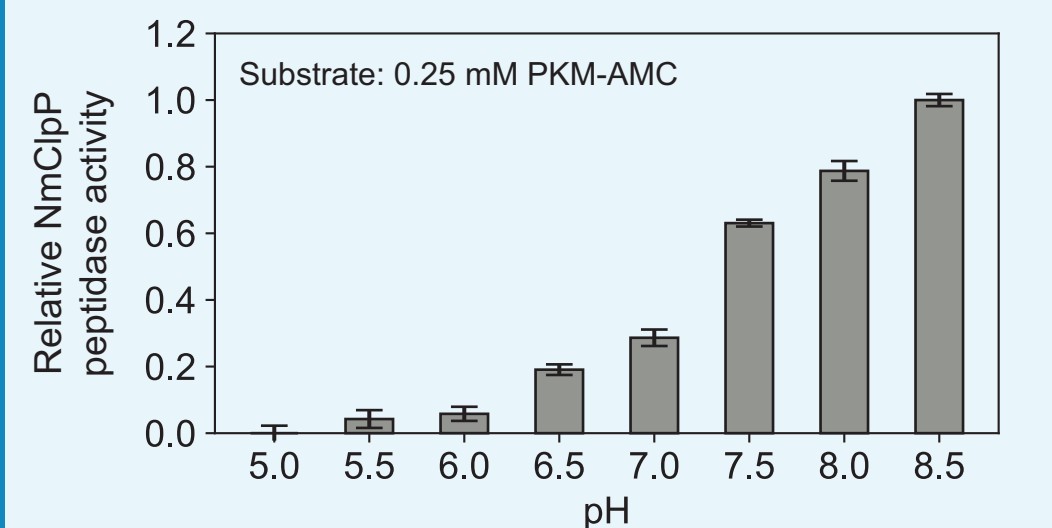

**Appendix 1—figure 1.** Peptidase rate measurements of NmClpP as a function of pH, monitored using the fluorogenic substrate PKM-AMC.

