## [Decision Letter]

**Acceptance summary:**

Bacterial ClpXP is an ATP-fueled chambered protease, akin to the 26S proteasome in eukaryotic cells. ClpXP consists of a hexameric AAA+ unfoldase (ClpX) that recognizes, unfolds, and threads SsrA-tagged substrates into the tetradecameric ClpP barrel composed of two heptameric rings, where substrates are degraded. Ripstein et al. now reports the cryo-EM structure of the ClpXP protease from *Neisseria meningitidis* engaged with the model substrate, GFP-SsrA. This study reveals two distinct conformations (termed A and B), at relatively high resolution. These structures are consistent with a sequential mechanism of substrate translocation, although the structures do not rule out other possibilities. Moreover, the structures suggest how a symmetry mismatch between ClpX and ClpP may contribute to the overall mechanism.

**Decision letter after peer review:**

Thank you for submitting your article "A processive rotary mechanism couples substrate unfolding and proteolysis in the ClpXP degradation machinery" for consideration by *eLife*. Your article has been reviewed by three peer reviewers, one of whom is a member of our Board of Reviewing Editors, and John Kuriyan as the Senior Editor. The following individual involved in review of your submission has agreed to reveal their identity: Gabriel C Lander (Reviewer #3).

The reviewers have discussed the reviews with one another and the Reviewing Editor has drafted this decision to help you prepare a revised submission.

Essential revisions:

1) It is mentioned that the E185Q Walker mutant was used for the structural studies to slow hydrolysis (Results paragraph one, but not mentioned in the Materials and methods). Was this mutant used for the functional studies (Figure 1) as well? In any case, how "slow" is this mutant? What fraction of the substrates will be unfolded within 10 mins, i.e., the time before flash freezing the samples?

2) All residues of substrate for both conformations (A and B) are modelled as alanine, but they do not clearly match the corresponding density maps. Please comment and discuss.

3) The lack of clear density for the sidechains of the substrate (or density of the substrate outside the ClpXP holoenzyme) for the two makes it difficult to uniquely correlate the two observed conformations of ClpXP with two different "translocation" states of the substrate. Although the model (Figure 5) is reasonable, the two structures presented in this study do not necessarily rule out other possibilities.

4) For conformation A, the resolution for the two promoters at either side of the seam is substantially lower (Figure 4—figure supplement 2) than the rest of the structure. Can one be sure that it is actually ADP bound to these protomers in conformation A? It could also be a resolution effect. Moreover, density for not only the nucleotide, but the surrounding regions should be included for all binding pockets.

5) Density for Mg^2+^ is mentioned, but the figure that is provided (Figure 4) shows the density at only one contour level, such that the putative Mg^2+^ density is merged with the density of the nucleotide. Two different contour levels should be shown. In any case, the "resolution" of the Mg^2+^ density does not seem to be as high as some of the sidechain densities shown and certainly not as high as estimated from the FSC. Please comment and discuss. Also, some the Mg^2+^ may not be properly modeled since they appear outside of density.

6) The present work represents the third cryoEM structure of the ClpXP protease and is preceded by the recent publication of *Listeria monocytogenes* ClpXP (Gatsogiannis et al., NSMB 2019). Contrary to the previous work, the structure by Ripstein et al. is of much higher resolution (2.3A vs ~4A), providing more accurate mechanistic insight. Perhaps most surprisingly, Gatsogiannis et al. reported the existence of unusual head-to-head ClpXP dimers that were the prevalent specimen in their cryoEM sample. Whether head-to-head dimers are artifactual or physiological is currently unclear. However, it seems appropriate to compare and comment on the previously published structures and to discern why such head-to-head dimers were not observed in the present study since full-length ClpX was also used here.

7) In the present structure, five of the six pore loops interact tightly with the substrate backbone supporting a processive substrate handover between neighboring subunits. This is somewhat unexpected as it is widely presumed that ClpX threads substrates stochastically. For instance, using a covalently linked ClpX hexamer, the Sauer and Baker groups showed that only one active ClpX subunit is sufficient for function. How can the present structures be reconciled with the wealth of biochemical and genetic data in the literature supporting a probabilistic mechanism?

8) To illuminate the structural basis for substrate engagement and threading, the authors used GFP-SsrA, which is more informative than the more commonly used casein that is natively unstructured. However, it is unclear how much of SsrA vs unfolded GFP is seen in the structure. Does GFP remain mostly folded and, if not, what is the structural/mechanical basis for GFP unfolding? The authors ought to comment on this.

9) The authors propose that the X5 subunit in conformation A contains an ATP molecule that is "primed" for hydrolysis. Lander et al. observed a similar nucleotide binding site environment in the Lon complex, and also proposed this scenario. However, this is incompatible with studies showing that an Arginine finger in trans is required for hydrolysis. The resolution of the reconstruction does not enable the authors to discern between ATP and a post-hydrolysis/long-lived ADP-Pi state that may be induced by the Walker B mutation. Unless the authors are able to posit a reasonable hydrolysis mechanism that explains how hydrolysis could occur in this chemical environment in a manner that satisfies prior observations regarding hydrolysis, this aspect of the mechanism must be reconsidered. This should include a more detailed and quantitative description of the Walker B motif becoming positioned "closer" to the ATP in this subunit.

10) The nominal resolution of the structure may enable a detailed description of the allostery involved in ClpP gate opening. It is mentioned that the rigidification is similar to that observed upon ADEP binding, but are there any differences observed or previously uncharacterized interactions? A more thorough description with associated figures should be included, as this aspect of the structure is particularly important.

11) The observed interaction between H230 within the RKH loop and substrate is intriguing and warrants further investigation. Despite all the mutagenesis that's been done on this loop, as far as I know no one has made a single point mutation at this histidine. The strength of the density between the H230 and substrate is indicative of a substantial interaction, the functional significance of which should be probed biochemically.

[Editors' note: further revisions were suggested prior to acceptance, as described below.]

Thank you for resubmitting your work entitled "A processive rotary mechanism couples substrate unfolding and proteolysis in the ClpXP degradation machinery" for further consideration by *eLife*. Your revised article has been evaluated favorably by John Kuriyan (Senior Editor) and a Reviewing Editor.

The manuscript has been improved but there are some remaining issues that need to be addressed before acceptance, as outlined below. Please address these issues in a revised manuscript, using your best judgement. We will make a final decision on the revised manuscript at the editorial level.

Reviewer #1:

I would like to thank the authors for addressing my concerns. I have no further comments and fully support publication in *eLife*.

Reviewer #2:

The manuscript by Ripstein et al. is substantially improved. The new data shown in Figure 5 addresses concerns regarding the proposed sequential mechanism, and the authors' comment regarding glutaraldehyde crosslinking used by Gatsogiannis et al., 2019 is adequate, although not further substantiated. Few minor issues remain that need clarification.

1) "The RKH loops are essential for the recognition of SsrA-tagged substrates by *E. coli* ClpX, while human ClpX, which lacks RKH loops loops but contains pore-1 loops with the same sequence as *E. coli* ClpX, fails to recognize substrates with the SsrA-tag."

Human ClpX features an RKL tripeptide (residues 401-403) instead of an "RKH motif" at the same position downstream of pore-1 loop. Since human ClpX fails to recognize ssrA-tagged substrates, it would seem that H230 is the main determinant for SsrA substrate binding as pointed out by one reviewer. While there is no reason to doubt the Farrell et al., 2007 findings, the explanation that electrostatic interactions via the charged Lys confer substrate specificity does not seem warranted according to the present structure. This statement needs to be revised.

2) Abstract: "… cyclical hydrolysis of ATP is coupled to concerted motions of ClpX loops…" This statement can be misleading, especially when the Abstract is read in isolation (e.g. in Pubmed), and should be revised. The authors' concluded that ATP hydrolysis occurs sequentially, but this was not stated in the Abstract. When taken in isolation, concerted motions refer to a third model distinct from the sequential and stochastic model.

Reviewer #3:

I am generally satisfied with the revised manuscript, and the authors have addressed all the points I brought up in my initial review. However, the authors introduce a new experiment (Figure 5), whose results they interpret as evidence of a sequential hydrolysis translocation mechanism. The experiment is nearly identical to one that was published as Figure 1 Martin et al., 2008 (PMID 18223658). The observed curve for GFP degradation has been previously established, and proposed to be unrelated to processivity of translocation due to the fast refolding of destabilized GFP intermediates. Lowering the motor speed enables GFP to refold and escape degradation, and could occur whether ClpX utilizes a probabilistic or sequential ATP hydrolysis mechanism. Further, given that the conformations associated with substrate engagement and commitment are unknown, and that these processing steps could also be perturbed by introducing a single slowly hydrolyzing subunit, one cannot exclude a probabilistic mechanism.

The manuscript provides many insights into the mechanism of ClpXP substrate processing and that this work will be well-received by the AAA community, but the conclusions regarding sequential hydrolysis are unfounded. I suggest that the experiments and interpretation associated with Figure 5 be removed.

---

## [Author Response]

Essential revisions:1) It is mentioned that the E185Q Walker mutant was used for the structural studies to slow hydrolysis (Results paragraph one, but not mentioned in the Materials and methods). Was this mutant used for the functional studies (Figure 1) as well? In any case, how "slow" is this mutant? What fraction of the substrates will be unfolded within 10 mins, i.e., the time before flash freezing the samples?

We apologize for the lack of clarity. The E185Q Walker B mutant was not used for the functional assays (Figure 1B). We have revised the figure caption to reflect this change:

“All measurements included GFP-SsrA and were performed in triplicate on WT ClpXP”

In a new figure, we show that the ATPase rate of the E185Q Walker B mutant of ClpX is ~17 times slower compared to the WT enzyme (Figure 5A). We believe that the substrate density in the maps corresponds to the unstructured region of the SsrA-tag of GFP that has been engaged and slowly translocated. Given the low activity, we believe the fraction of GFP that is unfolded prior to vitrification is negligible. We have clarified the text to reflect this:

“As the Walker B mutant has low unfolding activity (see below), the observed density likely corresponds to the unstructured region of the degron tag”

2) All residues of substrate for both conformations (A and B) are modelled as alanine, but they do not clearly match the corresponding density maps. Please comment and discuss.

While there is clear density for some bulkier side chains in the map, it was not possible to register the substrate peptide sequence in the density and therefore the substrate has been modeled as polyalanine. The density may correspond to the SsrA tag, although our attempts to fit that sequence into the density were unsuccessful, perhaps because the consensus structure is an average of multiple translocation steps of the SsrA sequence. We have added the following line in the Materials and methods to clarify this situation:

“While the experimental density for the substrate showed some bulky side chains, attempts to register the SsrA sequence in the density were unsuccessful and the substrate was modeled as polyalanine.”

3) The lack of clear density for the sidechains of the substrate (or density of the substrate outside the ClpXP holoenzyme) for the two makes it difficult to uniquely correlate the two observed conformations of ClpXP with two different "translocation" states of the substrate. Although the model (Figure 5) is reasonable, the two structures presented in this study do not necessarily rule out other possibilities.

We agree with the reviewers that a lack of register for the substrate chain complicates using it as a marker of translocation direction. However, in our proposed model the direction of substrate translocation is *towards* ClpP and is supported by both the nucleotide content of the observed states, and the pore loop positions. We have added new functional assays (Figure 5B – see reviewer comment #7) that further support this model. We do nonetheless accept that there are other possible models and have indicated as such in the text:

“The model described above is the simplest one that is consistent with our data. More complicated models can be envisioned that require additional states for which we have no experimental data.”

4) For conformation A, the resolution for the two promoters at either side of the seam is substantially lower (Figure 4—figure supplement 2) than the rest of the structure. Can one be sure that it is actually ADP bound to these protomers in conformation A? It could also be a resolution effect. Moreover, density for not only the nucleotide, but the surrounding regions should be included for all binding pockets.

While lower resolution near the seam in Conformation A makes the density for nucleotide in the binding pockets at the seam less clear, it is still sufficient to identify the nucleotide state. The ADP state of these pockets is also reinforced by the conformation of the surrounding residues. We have added a figure supplement (Figure 4—figure supplement 3) to show surrounding density as well:

“Figure 4—figure supplement 3. ATP binding pocket densities. Experimental density maps and models are shown for the ATP binding pockets of all protomers. Two numerical thresholds are shown to highlight differences in density.”

5) Density for Mg^2+^ is mentioned, but the figure that is provided (Figure 4) shows the density at only one contour level, such that the putative Mg^2+^ density is merged with the density of the nucleotide. Two different contour levels should be shown. In any case, the "resolution" of the Mg^2+^ density does not seem to be as high as some of the sidechain densities shown and certainly not as high as estimated from the FSC. Please comment and discuss. Also, some the Mg^2+^ may not be properly modeled since they appear outside of density.

We have added Figure 4—figure supplement 3 showing the two contour levels for putative Mg^2+^ (see reviewer comment #4). We believe that the Mg^2+^ density is comparable to other cryo-EM reconstructions at similar estimated resolutions. We have remodelled the Mg^2+^ such that it better corresponds to the observed density.

6) The present work represents the third cryoEM structure of the ClpXP protease and is preceded by the recent publication of Listeria monocytogenes ClpXP (Gatsogiannis et al., NSMB 2019). Contrary to the previous work, the structure by Ripstein et al. is of much higher resolution (2.3A vs ~4A), providing more accurate mechanistic insight. Perhaps most surprisingly, Gatsogiannis et al. reported the existence of unusual head-to-head ClpXP dimers that were the prevalent specimen in their cryoEM sample. Whether head-to-head dimers are artifactual or physiological is currently unclear. However, it seems appropriate to compare and comment on the previously published structures and to discern why such head-to-head dimers were not observed in the present study since full-length ClpX was also used here.

We did not observe any head-to-head dimers in any of our preparations. The authors of the *Listeria monocytogenes* ClpXPmanuscript utilized the glutaraldehyde chemical crosslinker in their sample preparation that may have affected the oligomeric state they observed. Our study did not use any crosslinking agent. We have added the following sentence to the Discussion:

“In the case of *L. monocytogenes* ClpXP, unusual head-to-head dimers were observed that appeared to be mediated by the zinc binding domains. Despite the presence of the zinc binding domains in our *N. meningitidis* construct, no such dimers were observed. While this difference may be due to differences across species, we note that the glutaraldehyde chemical crosslinker used to stabilize the *L. monocytogenes* ClpXP complex may have induced the formation of artefactual dimers.”

7) In the present structure, five of the six pore loops interact tightly with the substrate backbone supporting a processive substrate handover between neighboring subunits. This is somewhat unexpected as it is widely presumed that ClpX threads substrates stochastically. For instance, using a covalently linked ClpX hexamer, the Sauer and Baker groups showed that only one active ClpX subunit is sufficient for function. How can the present structures be reconciled with the wealth of biochemical and genetic data in the literature supporting a probabilistic mechanism?

In an effort to better understand the previously reported linked-hexamer data from the Sauer group in the context of our and many other substrate-bound AAA+ unfoldase structures that support a processive hand-over-hand mechanism, we performed our own mixed hexamer assays (Figure 5B). Here we mix WT and E185Q Walker B protomers of ClpX in different ratios and incubate them overnight so as to generate composite complexes with various protomer types whose exact composition and population can be calculated using combinatorial statistics. GFP-SsrA assays performed on samples described above show that the presence of a single E185Q Walker B protomer in the ClpX hexamer is enough to completely stop the degradation of GFP-SsrA by ClpXP. This further bolsters our processive model and is inconsistent with the previous stochastic model which predicts that unfolding and translocation is possible with just a single active subunit.

We would also note that in the experiments by the Sauer and Baker groups, the assumption is that the Walker B mutants are “dead”. As we show here (Figure 5A) and as reported by Sauer et. al elsewhere (Martin et al., 2008a), Walker B ClpX exhibits ~17 fold lower ATPase activity compared to the WT enzyme. As such the processivity of these mixed enzymes would be expected to be much lower than a fully WT complex. Furthermore, the substrates used in many of these experiments (e.g. Titin-SsrA, destabilized mutants of Titin-SsrA) have much lower stability compared to the tightly-folded GFP and may only require a few translocation steps to unfold. In the case of unstable or disordered substrates, translocation may take place slower compared to the WT enzyme but would register as “active”. Indeed, in our structures we show that ClpX binds GFP-SsrA (Figure 1—figure supplement 1) and believe that the substrate density in our maps corresponds to the unstructured SsrA tag of GFP which has been slowly translocated by our Walker B ClpX.

We have added the following text to the manuscript:

“The arrangement of nucleotides in our map suggests that substrate-bound ClpX hydrolyzes ATP in a sequential manner, contrary to previous reports that ATP hydrolysis by ClpX is a stochastic process (Martin et al., 2005). […] It is inconsistent with the previous stochastic model that predicts that unfolding and translocation are possible with just a single active subunit.”

And figure caption:

“Figure 5. A single defective ClpX protomer derails function. (A) ATPase rates for WT ClpX and the E185Q ClpX Walker B mutant. […] This observation contrasts the expected linear decrease (green line) for a stochastic model. (C) Schematic of the molecules produced via mixing.”

The associated Materials and methods section:

“…For the mixed protomer ClpX assays, WT ClpX and E185Q ClpX samples were purified separately as described above (i.e. without the addition of any nucleotide) and mixed at the indicated ratios. […] The wells included an ATP-regeneration system that contained 1.5 mM phosphoenolpyruvate, 0.2 mM NADH, 40 μg/mL pyruvate kinase, 40 μg/mL lactate dehydrogenase, and 2 mM MgATP at pH 8.2. Hydrolysis of ATP was monitored by loss of absorption at 340 nm with a Synergy Neo2 96-well microplate reader at 25 °C.”

8) To illuminate the structural basis for substrate engagement and threading, the authors used GFP-SsrA, which is more informative than the more commonly used casein that is natively unstructured. However, it is unclear how much of SsrA vs unfolded GFP is seen in the structure. Does GFP remain mostly folded and, if not, what is the structural/mechanical basis for GFP unfolding? The authors ought to comment on this.

Based on activity assays done with the E185Q Walker B mutant (see new Figure 5) we believe that GFP remains folded and the portion we observe in the structure corresponds to the C-terminal SsrA tag, which is natively unstructured and has begun to be translocated. We have clarified the text to reflect this.

“As the Walker B mutant has low unfolding activity, the density observed likely corresponds to the unstructured region of the degron tag.”

9) The authors propose that the X5 subunit in conformation A contains an ATP molecule that is "primed" for hydrolysis. Lander et al. observed a similar nucleotide binding site environment in the Lon complex, and also proposed this scenario. However, this is incompatible with studies showing that an Arginine finger in trans is required for hydrolysis. The resolution of the reconstruction does not enable the authors to discern between ATP and a post-hydrolysis/long-lived ADP-Pi state that may be induced by the Walker B mutation. Unless the authors are able to posit a reasonable hydrolysis mechanism that explains how hydrolysis could occur in this chemical environment in a manner that satisfies prior observations regarding hydrolysis, this aspect of the mechanism must be reconsidered. This should include a more detailed and quantitative description of the Walker B motif becoming positioned "closer" to the ATP in this subunit.

We agree with the reviewers on this point. It is entirely possible that the conformation we observed in this study is a stabilized post-hydrolysis state or indeed a mixture of pre- and post-hydrolysis states. While it is interesting that the sensor-2 arginine adopts a conformation similar to that of the canonical arginine finger, our maps are not at a sufficient resolution to adequately interpret a new mechanism and as such we have carefully reworded the text to remove mention of this site as being primed for hydrolysis and simply note that it is different from the other observed sites and may represent a post-hydrolysis/long-lived ADP-Pi state. We have clarified the text to reflect this:

“In the X5 protomer (at the bottom of the spiral), the adjacent X6 protomer and its arginine finger have pivoted away from the nucleotide into the LS position, allowing the sensor-II arginine to move closer to the γ-phosphate of the bound nucleotide. While there is clear density for the γ-phosphate, the resolution of our experimental map was not sufficient to differentiate between ATP and a long-lived post hydrolysis ADP/P_i_ state. Notably, this key sensor II-priming motif in the nucleotide site adjacent to an ADP bound site has also been reported for the Lon protease bound to substrate (Shin et al., 2019), a close relative in the HCLR clade of AAA+ proteins.”

10) The nominal resolution of the structure may enable a detailed description of the allostery involved in ClpP gate opening. It is mentioned that the rigidification is similar to that observed upon ADEP binding, but are there any differences observed or previously uncharacterized interactions? A more thorough description with associated figures should be included, as this aspect of the structure is particularly important.

We agree with the reviewers that this aspect of the structure is of particular interest. There are differences to the gate opening observed with ADEP compared with our ClpXP structures. Most notably not all gates assume the “up” conformation equally (Figure 2) and the offset of ClpX relative to ClpP creates new asymmetric interactions with the N-terminal gates. Further insights into the allosteric mechanism, particularly communication between ClpP protomers did not seem to differ from that of the high resolution ADEP structures. We believe Figure 2E and G and Figure 2—figure supplement 2 and the associated text of our original submission addressed this important point.

11) The observed interaction between H230 within the RKH loop and substrate is intriguing and warrants further investigation. Despite all the mutagenesis that's been done on this loop, as far as I know no one has made a single point mutation at this histidine. The strength of the density between the H230 and substrate is indicative of a substantial interaction, the functional significance of which should be probed biochemically.

We agree with the reviewers that the interaction of H230 and the substrate is intriguing, and that our models suggest it plays an important role in substrate processing. However, we do not think further mutagenesis work is warranted because RKH loop mutants have been extensively probed by the Sauer lab in the past (Farrell et al., 2007; Martin et al., 2008b) as well as in a recent preprint (Fei et al., 2019), including the single point mutation of the RKH histidine. These publications clearly show that mutating the RKH loop to AAA abrogates degradation of SsrA-tagged substrates, and furthermore that a simple swapping of the Lys and His positions (RKH to RHK) also results in a severe reduction of degradation of SsrA-tagged substrates. Additionally the single point mutation of the histidine to alanine (RKA) resulted in loss of degradation for both SsrA and λO tagged substrates (Farrell et al., 2007). The specific positioning of a single H230 in our structure may provide an explanation for the sensitivity of activity to changes in this loop. We have added the following text and citations to discuss this point:

“The differences in these interactions may reflect the fact that while pore-1 loop and RKH loop motifs are critical for forming interactions with substrate, and in the case of pore-1 loops in particular, in providing the force for translocation, additional contacts may differ between species and may be necessary for recognition of degradation signals and for substrate specificity. […] The tight contact between H230 and the substrate in our maps provides a structural basis for understanding these mutagenesis results.”

[Editors' note: further revisions were suggested prior to acceptance, as described below.]

Reviewer #2:

The manuscript by Ripstein et al. is substantially improved. The new data shown in Figure 5 addresses concerns regarding the proposed sequential mechanism, and the authors' comment regarding glutaraldehyde crosslinking used by Gatsogiannis et al., 2019 is adequate, although not further substantiated. Few minor issues remain that need clarification.1) "The RKH loops are essential for the recognition of SsrA-tagged substrates by *E. coli* ClpX, while human ClpX, which lacks RKH loops loops but contains pore-1 loops with the same sequence as *E. coli* ClpX, fails to recognize substrates with the SsrA-tag."Human ClpX features an RKL tripeptide (residues 401-403) instead of an "RKH motif" at the same position downstream of pore-1 loop. Since human ClpX fails to recognize ssrA-tagged substrates, it would seem that H230 is the main determinant for SsrA substrate binding as pointed out by one reviewer. While there is no reason to doubt the Farrell et al., 2007 findings, the explanation that electrostatic interactions via the charged Lys confer substrate specificity does not seem warranted according to the present structure. This statement needs to be revised.

We thank the reviewer for brinting this point to our attention. We have removed the text corresponding to the electrostatics explanation and now mention the mitchondrial ClpX RKL tripeptide:

“The RKH loops are essential for the recognition of SsrA-tagged substrates by *E. coli* ClpX, while human ClpX, which substitutes the RKH sequence with an RKL tripeptide and contains pore-1 loops with the same sequence as *E. coli* ClpX, fails to recognize substrates with the SsrA-tag..”

2) Abstract: "… cyclical hydrolysis of ATP is coupled to concerted motions of ClpX loops…" This statement can be misleading, especially when the Abstract is read in isolation (e.g. in Pubmed), and should be revised. The authors' concluded that ATP hydrolysis occurs sequentially, but this was not stated in the Abstract. When taken in isolation, concerted motions refer to a third model distinct from the sequential and stochastic model.

The text has been modified as follows to address this ambiguity:

“The structures allow development of a model in which the sequential hydrolysis of ATP is coupled to motions of ClpX loops that lead to directional substrate translocation and ClpX rotation relative to ClpP.”

Reviewer #3:

I am generally satisfied with the revised manuscript, and the authors have addressed all the points I brought up in my initial review. However, the authors introduce a new experiment (Figure 5), whose results they interpret as evidence of a sequential hydrolysis translocation mechanism. The experiment is nearly identical to one that was published as Figure 1 Martin et al., 2008 (PMID 18223658). The observed curve for GFP degradation has been previously established, and proposed to be unrelated to processivity of translocation due to the fast refolding of destabilized GFP intermediates. Lowering the motor speed enables GFP to refold and escape degradation, and could occur whether ClpX utilizes a probabilistic or sequential ATP hydrolysis mechanism. Further, given that the conformations associated with substrate engagement and commitment are unknown, and that these processing steps could also be perturbed by introducing a single slowly hydrolyzing subunit, one cannot exclude a probabilistic mechanism.The manuscript provides many insights into the mechanism of ClpXP substrate processing and that this work will be well-received by the AAA community, but the conclusions regarding sequential hydrolysis are unfounded. I suggest that the experiments and interpretation associated with Figure 5 be removed.

We thank this reviewer for his/her comments. The sequential vs. stochastic discussion is certainly controversial and merits a more detailed investigation. As per this reviewers’ suggestion, we have removed Figure 5 and all the associated text. We have added the following to the Discussion:

“The model described above is the simplest one that is consistent with our data. While our structures suggest that a sequential hydrolysis model exists for ClpXP, more complicated models can be envisioned that require additional states for which we have no experimental data. Indeed biochemical evidence suggests the possibility that when one or more subunits are catalytically deficient a more complex mechanism may be operational, allowing ClpXP to deviate from a strictly sequential hydrolysis scheme (Martin et al., 2008b).”